# Representation and Improved Parameterization of Reservoir Operation in Hydrological and Land Surface Models

Fuad Yassin[1], Saman Razavi[1], Mohamed Elshamy[1], Bruce Davison[2], Gonzalo Sapriza-Azuri[3], Howard Wheater[1]

[1]Global Institute for Water Security, University of Saskatchewan, National Hydrology Research Centre, 11 Innovation Boulevard, Saskatoon, SK, S7N 2H5, Canada
[2]National Hydrology Research Center, Environment Canada, 11 Innovation Boulevard, Saskatoon, SK, S7N 3H5, Canada
[3]Departamento del Agua, Centro Universitario Región Litoral Norte, Universidad de la República del Uruguay, Salto, Uruguay

*Correspondence to*: Fuad Yassin (fuad.yassin@usask.ca)

**Abstract.** Reservoirs significantly affect flow regimes in watershed systems by changing the magnitude and timing of streamflows. Failure to represent these effects limits the performance of hydrological and land surface models (H-LSMs) in the many highly regulated basins across the globe and limits the applicability of such models to investigate the futures of watershed systems through scenario analysis (e.g., scenarios of climate, land use, or reservoir regulation changes). An adequate representation of reservoirs and their operation in an H-LSM is therefore essential for a realistic representation of the downstream flow regime. In this paper, we present a general parametric reservoir operation model based on piecewise linear relationships between reservoir storage, inflow, and release, to approximate actual reservoir operations. For the identification of the model parameters, we propose two strategies: (a) a "generalized" parameterization that requires a relatively limited amount of data; and (b) direct calibration via multi-objective optimization when more data on historical storage and release are available. We use data from 37 reservoir case studies located in several regions across the globe for developing and testing the model. We further build this reservoir operation model into the MESH modelling system, which is a large-scale H-LSM. Our results across the case studies show that the proposed reservoir model with both of the parameter identification strategies leads to improved simulation accuracy compared with the other widely used approaches for reservoir operation simulation. We further show the significance of enabling MESH with this reservoir model and discuss the interdependent effects of the simulation accuracy of natural processes and that of reservoir operation on the overall model performance. The reservoir operation model is generic and can be integrated into any H-LSM.

## 1 Introduction

### 1.1 Background and Motivation

Human interventions in natural hydrologic systems, through damming and storing water, diversion, surface and groundwater abstraction, irrigation, and land use change, have significantly altered the natural river flow regimes and the terrestrial water cycle of many river basins (Vörösmarty et al., 1997, 2003; Oki and Kanae, 2006; Wisser et al., 2010;

Haddeland et al., 2014; Biemans et al., 2011). These interventions are to fulfil different types of demands such as domestic, industrial, irrigation, and hydropower demands, and to meet other needs such as flood control and conservation of aquatic habitats. With a total storage volume of more than 8000 km$^3$ (ICOLD, 2003; Vörösmarty et al., 2003; Hanasaki et al., 2006), more than 50,000 dams have been constructed globally to regulate more than half of the world's large river systems (Nilsson et al., 2005). The aggregate storage volume of these dams is greater than 20% of the global mean annual runoff (Vörösmarty et al., 1997) and is three times the annual average water storage in world's river channels (Hanasaki et al., 2006).

Despite the benefits in terms of enhancing water availability in support of food security, power supply, etc., dams result in several negative environmental and social consequences. Adverse environmental effects include changes in natural river dynamics in terms of water temperature, sediment and nutrient transport, etc. and the fragmentation and loss of biodiversity (Vörösmarty et al., 2010). Reservoirs can also intensify evaporation, by increasing the surface area of water exposed to direct sunlight and air, and through water supply for irrigation (de Rosnay et al., 2003; Pokhrel et al., 2012). Other environmental impacts of dams include the alteration of landscape due to dam construction and changes to land-atmosphere interaction that can have a profound impact on local/regional climate (Hossain et al., 2012; Degu et al., 2011). Adverse social effects include the displacement of people living near the dam site, changes to fishing patterns, and downstream erosion (Strobl and Strobl, 2011, p. 449). There are research gaps remaining in evaluating both positive and negative social impacts of dams (Kirchherr et al., 2016). Such gaps have been the subject of many studies in both academia and industry for years, and recently, have led to the formalization of this study area of "socio-hydrology" (Sivapalan et al., 2012; Sivakumar, 2012).

Dams and reservoirs change the natural flow regimes in rivers, both in terms of magnitude and timing of flows. As a result, for rivers that contain large or small dams and reservoirs, flow regimes are a combination of natural and managed flows. Various modeling communities manage this mix of natural and managed flows differently. Archfield et al. (2015) compare three families of models that can be used at continental scales: catchment models (CM), global water security models (GWSM), and land-surface models (LSM). CMs generally ignore water management and focus on unmanaged headwater catchments. GWSMs have been utilized in global-scale streamflow simulations and generally focus on large-scale water management issues, which are made difficult by a lack of data on large-scale water management and operational decisions. LSMs have traditionally focused on providing lower boundary conditions for atmospheric models, but are increasingly being used for hydrological applications in which they are referred to as Hydrologic Land Surface Models (H-LSMs). LSMs generally ignore water management (Clark et al., 2015; Davison et al., 2016), with a few exceptions (e.g., Voisin et al. 2013a, 2013b). A fourth family of water models, that is relevant to the work presented here, are water management models (WMM) (Labadie, 1995; Yates et al., 2005). Water modellers who know how the water is managed within their basins of interest generally use WMMs (Lund and Guzman, 1999; Labadie, 2004; Kasprzyk et al., 2013). These models contain very detailed representations of water management decisions, but often consider natural flow processes in a much more rudimentary fashion than CMs.

Modeling the many managed basins around the world using the current generation of CMs or LSMs can result in models with limited fidelity and question the credibility of their predictions of future water resources in basins with dams and reservoirs. Therefore, there is a pressing need for better characterization and integration of the operation of dams and reservoirs

into hydrological modelling frameworks using CMs and LSMs (Nazemi and Wheater, 2015a and 2015b; Pokhrel et al., 2016; Wada et al., 2017). This need motivated the objectives of this study, described in Section 1.2, and some previous research, outlined in Section 2. The integration of reservoir regulation into hydrological modeling frameworks will improve our ability to simulate highly regulated basins around the globe, leading to better understanding of historical conditions of water resource systems and improved assessment and prediction of their future vulnerability to climate and environmental change.

## 1.2 Objectives

Building upon previous research, this study aims to:

- Develop and test an improved reservoir operation model that can be integrated into any CM and LSM at any scale, but in particular at large scales. Of interest is a simple but effective parametrization that can be adjusted to varying levels of data availability.
- Integrate the developed reservoir operation model into an LSM and evaluate its performance when working in combination of other processes in the model. Also of interest is to assess the potential conceptual and technical issues in this integration.

. Another potentially very fruitful, but largely unexplored approach would be to couple CMs and LSMs with WMMs, but that approach is not examined here due to the fact that WMMs generally require extensive information on how water is managed within a basin, whereas we are particularly interested in the more generic case when this information is likely to be limited or unavailable.

The organization of the remainder of the paper is as follows. Section 2 reviews different existing approaches in the literature for the representation of reservoir operation in hydrologic models. Section 3 presents the proposed reservoir operation model and the metrics used to evaluate it, in comparison with other existing models. Section 4 provides a description of the reservoir dataset used for the developments and testing. Section 5 presents the assessment results and comparisons. Section 6 ends the paper with a summary of the main findings and conclusions.

## 2 Existing Reservoir Models in Catchment Models and Land Surface Models

An adequate representation of human interventions in Earth systems models is a major challenge. Systematic approaches towards full integration are needed as outlined in the recent studies of Nazemi and Wheater (2015a and 2015b), Wada et al. (2017), and Pokhrel et al. (2016). In this work, our focus is on the representation of dam and reservoir operation in catchment models (CMs) and Land Surface Models (LSMs), particularly when used at large scales. While there has been tremendous progress in the last decades in modelling the operation and management of reservoir systems at local to regional scales (e.g., Castelletti et al., 2010; Chang et al., 2010; Fraternali et al., 2012; Razavi et al. 2012; Asadzadeh et al. 2013; Guo et al., 2013), a gap still exists between the methodologies applied for local/regional-scale reservoir operation and management and the representation of reservoir operation in Earth systems models, particularly in LSMs. This gap is due to a two-fold

challenge. First, the upscaling of methodologies used at smaller scales to larger scales is non-trivial; and second, the availability of data on reservoir operation and water use is often limited in many parts of the world. For example, the reservoir purpose and operational details are not always known and large reservoirs typically serve several purposes (Wisser et al., 2010). As a result, most current hydrological modeling activities with CMs and LSMs, if not all, offer only a limited capability in simulating reservoir operations, whereas reservoir operation in practice involves a complex set of human-driven processes and decisions.

The existing reservoir operation methods in hydrologic models can be categorized roughly into three groups, based on their level of complexity in representing flow regulation; (I) natural lake methods, (II) inflow-and-demand based methods, III) artificial neural network techniques, and IVII) target storage-and-release based methods.

## 2.1 Natural lake methods

The most primitive methods use formulations developed for the simulation of natural lakes or uncontrolled reservoirs. In these methods, the downstream release is calculated as a function of reservoir storage characterized by some empirical parameters (Meigh et al., 1999; Döll et al., 2003; Pietroniro et al., 2007; Rost et al., 2008). For instance, Meigh et al. (1999) calculate the release by $Q_t = S_t^{1.5}$ where $Q_t$ and $S_t$ are release and reservoir storage, respectively. Their method was later modified by Döll et al. (2003) such that $Q_t = b_1 (S_t - S_{min}) (\frac{S_t - S_{min}}{S_{max} - S_{min}})^{b_2}$ where $b_1$ and $b_2$ are release coefficients, and $S_{min}$ and $S_{max}$ are minimum and maximum allowable reservoir storages. The advantage of this method, as shown in Döll et al. (2003), is its minimal data requirement, which supports its global applicability to model lakes, reservoirs and wetlands. However, it has limited functionality to adequately represent managed reservoirs due to not accounting for reservoir operation policies to constrain or increase releases at different phases of reservoir storage dynamics. Such simplistic methods ignore the fact that the operation of a reservoir depends on the reservoir purpose and the seasonal pattern of the mismatch between the demands it supports and the inflow it receives.

## 2.2 Inflow-and-demand based methods

The inflow-and-demand based methods include reservoir water balance models that determine reservoir release using a function that accounts for inflow or a combination of inflow and demands. The simplified method in this group is the method used in Wisser et al. (2010), it estimates the release as a function of mean annual inflow and a set of empirical parameters that can be calibrated in the absence of information on the actual operation of a reservoir.

Hanasaki et al. (2006) pioneered the development of inflow-and-demand reservoir models and laid the foundation for many subsequent developments. The method of Hanasaki et al. (2006) simulates reservoir release at a monthly time step within a global routing model, and accounts for water withdrawals for reservoirs categorized as irrigation reservoirs. They grouped reservoirs serving all others purposes as non-irrigation reservoirs. This approach first estimates a provisional total annual

release at the beginning of the water year based on the long-term mean annual inflow adjusted by an annual release coefficient. Then, a monthly provisional release is estimated based on the purpose of the reservoir (irrigation or non-irrigation). Downstream demands are accounted for in irrigation reservoirs only. The provisional monthly release for large reservoirs is then modified by the annual release coefficient to calculate the actual monthly release, and the provisional monthly release for small reservoirs is additionally adjusted based on the monthly inflow to calculate the actual monthly release. The release coefficient is estimated as a function of the reservoir storage at the beginning of the operational year and the reservoir capacity (the formulation of Hanasaki et al. (2006) is briefly explained in section 3.4). The release coefficient reduces the current year release if the storage at the beginning is low and vice-versa. Thus, the release coefficient accounts for inter-annual variability and facilitates the representation of strategies to overcome reservoir depletion in dry years and flood overtopping in wet years.

The implementation of the release coefficient is one of the limitations of Hanasaki et al. (2006), because it depends only on the year's initial storage and does not account for the actual inflow of the current operational year, i.e. it does not use foresight. The initial storage reflects the recent past of the operation of the reservoir, while the actual inflow could be considerably different than the long-term mean annual inflow. For instance a sequence of low flow years would result in a low initial storage while the current year inflow (which is not known yet) could be high, and vice versa. Additionally the simplification of complex reservoir operation in Hanasaki et al. (2006) by using the mean annual inflow and a release constraining coefficient produces errors. However, the method is generic and has low data requirements which are advantageous. The results showed that the reservoir algorithm improved monthly discharge simulation compared to the natural lake method (Hanasaki et al., 2006). The approach is effective and has found wide applicability in several global hydrological and land surface models.

The original Hanasaki et al. (2006) reservoir model has been modified in subsequent studies to address some of it limitations. For example, it has been modified for water extraction and other reservoir functions such as fulfilling environmental flows (Hanasaki et al., 2008a, b; Pokhrel et al., 2012a), and been adjusted to address direct precipitation over and evaporation from the reservoir (Döll et al., 2009).

Biemans et al. (2011) added new functionalities to the Hanasaki et al. (2006) reservoir model related to irrigation water demand and supply distribution and ran it at a daily time step. Their contributions include: 1) modifying irrigation withdrawals to account for conveyance losses and irrigation efficiency, 2) adjusting the minimum release to 10% of the mean monthly inflow, 3) prioritizing irrigation over flood control, 4) using regulated flow instead of natural flow, to estimate mean annual inflow, 5) storing the "flow to be released" for five days in the reservoir – to mimic the storage within the conveyance system – before it is released to the river. Voisin et al. (2013a) further modified the reservoir model of Hanasaki et al. (2006) to include multipurpose functionalities (irrigation and flood control) by changing the operation to release more before the onset of snowmelt-flood season so that there will be sufficient room to store flood waters form snowmelt in the reservoir. The modification requires the specification of a flood control period. Voisin et al. (2013a) have also evaluated the uncertainty of reservoir simulation by comparing withdrawal vs. consumptive demands, and natural vs. regulated flow for configuring operating rules. The results of Voisin et al. (2013a) demonstrated that adding flood control in reservoir operation, along with

a parametrization using mean annual natural inflow, and mean monthly withdrawals, improves the reservoir storage and flow simulation.

Haddeland et al. (2006) developed another pioneering generic reservoir model that has been implemented in a routing model at a daily time step to study the impact of reservoir and irrigation water withdrawals on continental surface water fluxes. The model is retrospective, i.e., it assumes full knowledge of the upcoming operation year reservoir inflow. The reservoir operation is conducted using an optimization scheme to determine the optimal release to satisfy different sectoral demands and targets that are defined in the form of objective functions. In the case of a multipurpose reservoir, the model gives priority to irrigation demand, followed by flood control and hydropower production. Minimum flow is estimated using natural flow based on seven-day consecutive low flows with a ten year recurrence period. The flood protection objective function is minimizing reservoir release above the bankfull discharge, which is estimated using the long-term mean of annual maximum discharge. Irrigation is optimized to satisfy downstream irrigation demand, while hydropower is optimized to increase power production. Predicting inflows for the current operational year, if possible, would allow the method to optimize the release while accounting for the whole operational year, otherwise to optimize day to day release without accounting for the remaining operational year would require several constraints. The maximum daily release is set based on the reservoir water balance that sets the storage at the end of the operational year to vary between 60 to 80% of the maximum capacity.

Similar to Hanasaki et al. (2006), the model of Haddeland et al. (2006) is favourable due to its generic formulation and capability to operate multipurpose reservoirs, and to extract water for irrigation from reservoir. These make the model applicable for large-scale hydrologic models, when data on operational policies are limited (Adam et al., 2007; Van Beek et al., 2011). One limitation of Haddeland et al., (2006) is that it requires knowledge of the future inflow for each reservoir so that the optimization can be conducted to determine the optimal release. Another limitation is that the release can deviate from the actual value because of simplifications of the objective function and errors from irrigation demand calculation. The algorithm does not represent reservoirs with multi-year operational policies (Adam et al., 2007) and also requires to run the model many times to optimize the reservoir release.

Adam et al. (2007) modified the Haddeland et al. (2006) reservoir model parameterization to include: 1) estimated minimum flow based on observed mean winter flow, 2) reservoir filling phase, 3) storage-area-depth relationship following the regular shape approximation of Liebe et al. (2005), 4) a seasonally varying hydropower production economic value that can be calibrated for hydropower production instead of a constant one. van Beek et al. (2011) further modified the retrospective inflow assumption to prospective model by approximating the upcoming operational year inflow based on previous years' inflow (requires historical inflow observation) and then adjusting the release and demand every month using the actual inflow as estimated from a hydrologic model.

Solander et al. (2015) tested and compared six generic equations to represent reservoir release and storage simulations. The complexity of equations tested varies from the simplest case that assumes reservoir outflow equals inflow (no-reservoir assumption), to a more complex representation using separate linear functions during reservoir filling and release periods. While the reservoir filling and release seasons were identified using long-term mean temperature, their respective release

equations are configured as a function of reservoir inflow, storage, and optimized seasonal empirical parameters. Their results on California reservoirs showed that the equation dependent on inflow is best for recharge season, while release during the drawdown season was better represented as a function of storage. Despite failing for highly regulated reservoirs, their study demonstrated the possibility of generalizing the seasonal empirical parameters as a function of the ratio between winter inflows to storage capacity. However, further testing is required to examine the usefulness of Solander et al. (2015)'s in different region, such as cold regions, with different filling and release seasonality.

Although the inflow-and-demand based models provide improved results compared to the natural lake approach, these models do not accurately reproduce observed flows (Adam et al, 2007; Haddeland et al., 2006; Coerver et al., 2018). Overall, while the above methods have better flexibility for coupling with global hydrological and land surface models, the methods have limitations in accounting for details of reservoir operation. For an adequate representation of reservoirs, particularly multi-purpose reservoirs and/or those with multi-year carry-over capacity, it is important to consider *reservoir zoning* and adjust reservoir release formulations for different storage levels. The absence of this consideration may limit the capability of this group of methods in representing complex reservoir operations.

## 2.3 Neural network-based methods

Artificial neural network (NN) models have been applied to establish data-driven rules that relate reservoir storage, inflow and release data. This type of models (1) extensive data on reservoir release, storage, inflow, but minimal prior expert knowledge of the reservoir operation, and (2) extensive training of a model for each individual reservoir to deduce the reservoir operation rules. Neural network techniques have been widely used beyond reservoir operation applications (e.g., flood forecasting, streamflow simulation, water quality (Maier and Dandy, 2001; Razavi and Karamouz, 2007)) and more recently has shown promise in reproducing historical reservoir operations (Coerver et al., 2018).

The study of Coerver et al. (2018) provides detailed background on NN applications for deduction of reservoir operation rules, and also demonstrates the performance of NN-based fuzzy rules to describe the reservoir release decisions. The analysis of Coerver et al. (2018) involves different levels of input complexity for the neural network setups, such as the importance of accounting for inflow prediction and time of the season on the reservoir operation performance. Another similar application was shown by Ehsani et al. (2016) who demonstrated a general reservoir operation scheme that uses an NN technique to map the general input/output relationships to actual operating rules of seventeen dams. Ehsani et al. (2016) demonstrated the possibility of aggregating multiple reservoirs that are closely located, so that their integrated effect can be accounted for in large-scale hydrological modeling studies. In a subsequent study, Ehsani et al. (2017) integrated the reservoir model of Ehsani et al. (2016) into a global water security model to study reservoir operations under climate change.

While these studies demonstrated that, the NN-based models can reproduce historical reservoir operation data and possibly outperform the widely used reservoir simulation models such as those of Hanasaki et al. (2006) and Wisser et al. (2010), the user of such models may have to deal with a fundamental limitation, i.e., their "black-box" nature. This limits their ability to provide insight into the underlying mechanisms of reservoir operation, and might masks possible shortcomings in a

derived NN model. Further, the credibility of their performance in extrapolation beyond the historical data can be in question, as they fully ignore the expert knowledge available on the actual physical and socio-economic processes that govern reservoir operations. Together, these limit the interpretation of results and their applicability in a changing environment. There has been some recent research efforts to reformulate neural networks such that they can overcome these limitations (e.g., see Razavi and Tolson, 2011).

## 2.4 Target storage-and-release based methods

The target storage-and-release based methods aim to emulate actual rule curves (i.e., reservoir target storage and release for different times of the year) that guide reservoir operators to decide on downstream releases (Burek et al., 2013; Yates et al., 2005; Neitsch et al., 2005). The target levels of storage divide the total reservoir storage capacity into multiple zones. For example, in the SWAT model (Arnold et al., 1998), a reservoir model is available in which the total storage of a reservoir is divided into sediment, principal, flood control, and emergency flood control zones where each zone is either specified by the user or as a function of soil moisture wetness (Neitsch et al., 2005). Wu and Chen (2012) modified this approach by changing the reservoir zoning model and developed a reservoir release simulation strategy that uses a decision-based parameterization to better fit both storage and release of multi-purpose reservoirs. However, they reported only one application of this strategy to a local-scale reservoir, and its comprehensive evaluation needs to be performed on other reservoirs in other regions with different climates, levels of regulation, and allocation objectives.

Zhao et al. (2016) integrated a reservoir regulation module into a hydrology model, requiring user-specified (based on observed data) values to divide the reservoir into inactive, conservation and flood control zones. In their module, the release from the conservation zone is determined using water demand, which includes multi-sectorial demand and environmental flow. The release from the flood storage zone is decided as a function of inflow (classified as flood inflow or non-flood inflow), downstream channel current discharge and downstream maximum discharge. At the time of flood, if the downstream discharge is below the maximum limit, release from flood storage zone is estimated using available storage above conservation zone, multiplied by a weight parameter which allows to release more water. If the downstream discharge is at maximum capacity there is no release from flood storage zone. Finally, any storage above the flood storage zone is automatically released. Additionally, Zhao et al. (2016) added the possibility to operate reservoirs conjunctively by giving release priorities to immediate downstream demands and by limiting the release if the downstream reservoir is within flood storage zone. The results of reservoir integration showed improved capability of the hydrological model to simulate storage and release for Lake Whitney and Auilla Lake in Texas. The limitation of Zhao et al. (2016) for wider application is that there is no generic formulation of reservoir zoning (requires user specification) and evaluation was only performed on two reservoirs.

Similarly, Burek et al. (2013) divided the total reservoir storage into conservative, normal and flood zones within the LISFLOOD model, and defined release in accordance with these storage zones using multiple-linear regression. Zajac et al. (2017) showed the applicability of this method to capture the effects of lakes and reservoirs globally using a parameterization that depends on naturalized inflow and maximum storage. Their results showed that the inclusion of reservoirs and lakes in a

hydrologic model through this method helped improve streamflow simulation for many stations, but the performance in replicating observed storage dynamics was not reported.

Overall, the primary advantage of methods in this category is that they allow approximation of reservoir-release policy and have the potential of making use of detailed data on a reservoir when available. Their main limitation, however, is their relatively high data demands. When data are available, methods under this category have the potential to enhance the representation of dams and reservoirs in terms of both reservoir storage and release, while adapting to the seasonality and change in operations on different time scales from daily to seasonal. These methods seem advantageous to NN-based models as their functioning are transparent, accounting for the governing processes, while requiring similar data.

Given the advances in the field and the growing availability of data sources, the target storage-and-release methods seem to be the most promising, as they can better simulate the reservoir operation dynamics (the dynamics of both storage and release). The data requirement includes data on observed inflow, observed release, observed storage (level) and reservoir physical characteristics. Reservoir level data are available for most lakes and reservoirs in the public domain, particularly in North America. These data can be converted to reservoir storage using reservoir elevation-area-volume relationships or by using area-volume relationships approximated by regular geometric shapes (Yigzaw et al., 2018; Liebe et al., 2005; Lehner et al., 2011). Inflows to and releases from a reservoir can be approximated by streamflow stations located upstream and downstream of the reservoir, respectively. Further, satellite missions such as MODIS (Savtchenko et al., 2004) and satellite radar altimetry are providing information on lake and reservoir surface area dynamics and reservoir water elevation for some large reservoirs. The combination of MODIS and satellite radar altimetry allows to derive storage-area-depth relationships (Gao et al., 2012; Andreadis et al., 2007; Zhang et al., 2014; Yoon and Beighley, 2015). The planned SWOT (2021) mission (Garambois and Monnier, 2015) will increase the availability of water level data for smaller rivers (with widths going down to 100m) that can be potentially converted to discharge to estimate reservoir inflows and downstream reservoir releases.

## 3 Material & Methods

This study aimed to develop an improved reservoir model that better emulates reservoir operation for large-scale hydrologic modelling application in terms of both reservoir storage and release, following the previous advances in target-storage-and release-based methods reviewed in Section 2.4. In this section, we present the characteristics and formulation of our reservoir model. The reservoir water balance is maintained using the continuity equation, as shown in finite difference form in Equation 1. The aim is to estimate unknown storage $S_t$ and release $Q_t$ at the current time step based on the storage at the previous time step $S_{t-1}$ and precipitation (P) over the reservoir, evaporation (E) from the reservoir and inflows (I) during the current time step. When integrated within an H-LSM model, the inflow will be the modelled value of the upstream catchment that would account for delays in the precipitation-runoff generation and routing. This equation is solved in conjunction with the parametrization equations presented in the next section for reservoir releases to compute $S_t$ and $Q_t$.

$$\frac{S_t - S_{t-1}}{\Delta t} = \frac{I_t + I_{t-1}}{2} - \frac{Q_t + Q_{t-1}}{2} + \frac{P_t + P_{t-1}}{2} - \frac{E_t + E_{t-1}}{2} \qquad (1)$$

### 3.1 Proposed Reservoir Operation Model

A detailed description of our proposed target storage-and-release model (or target-release model for brevity) is provided here. This model is formulated in the form of parametric piecewise linear functions that approximate the reservoir

release rules that may be used by reservoir operators. This model can be set up on any time scale; in the case studies reported here, we define the target levels to *dynamically* change over time. We call the model the "Dynamically Zoned Target Release (DZTR)" Model. Piecewise linear function-based reservoir operation models have already been used to solve complex reservoir operation and water resources management problems (e.g., Razavi et al., 2013; Asadzadeh et al., 2014). A systematic integration of such models into large-scale hydrological modeling has been reported in Burek et al. (2013) as implemented in

the LISFLOOD hydrological model, and in Neitsch et al. (2005) as implemented in the SWAT model. Our DZTR model is a generalization of the method developed by Razavi et al. (2013), which may also be viewed as a modification to the model proposed by Burek et al. (2013) in terms of parametrization and reservoir zoning. Fig. 1 shows the schematic representation of DZTR; Fig. 1a shows the reservoir zoning and Fig. 1b shows the piecewise-linear functions to estimate the release for each zone based on DZTR.

The DZTR model divides reservoir storage into five zones in a similar fashion to Wu and Chen (2012) and Burek et al. (2013), namely dead storage, critical storage, normal storage, flood storage, and emergency storage. Whenever storage is below the emergency storage zone, release only occurs through the bottom outlet, but when the storage is within that zone, release happens through both of bottom outlet and the spillway. In the absence of data, the dead storage (Zone 0) is assumed to be 10% of the maximum storage after Döll et al. (2009). To estimate the remaining storage zones in cases where no

operational information on a reservoir is available, we propose two alternative strategies: (1) setting the zones based on suggested exceedance probabilities on historical reservoir storage time series, (2) optimizing these zones to reproduce the observed storage and release time series. Target releases for each zone can be obtained in a similar fashion. These target storages and releases are allowed to vary each month (or on any other arbitrarily selected time resolution) to allow a better representation of the seasonality of reservoir operation.

When reservoir storage is within the dead storage zone (Zone 0), the reservoir release is zero (equation 3). In Zone 1 (critical storage zone), the reservoir release is a function of storage at a given time step and the critical release target value (equation 4). In this zone, the reservoir operates to avoid storage depletion while trying to support environmental flow requirements defined as a critical (or minimum) release. In Zone 2 (normal storage zone), the reservoir release is purely governed by reservoir storage and varies between critical and normal release targets (equation 5). In this zone, the downstream

release is greater for higher levels of storage. In Zone 3, the release decision considers both reservoir storage and inflow in that time step as well as the normal and maximum release targets (Equation 6). When in this zone, two scenarios may occur: (A) the amount of inflow in a time step is equal to or less than the normal release rate; (B) the amount of inflow in this time

step is greater than the normal release rate. As formulated in Equation 6, in the case of scenario B, the inflow rate comes to play to augment the release in an attempt to keep the reservoir level within the normal storage zone. Scenario B is expected to occur more frequently in smaller reservoirs that only have "within-year" storage capacity, while scenario A should be more commonly seen with larger reservoirs that have "multi-year" carry over capacity. Hanasaki et al. (2006) suggested that

reservoirs that have a ratio of storage capacity to mean annual inflow (referred to as $c$) of less than 0.5 be assumed as within-year reservoirs and the ones with a ratio of 0.5 and above be considered as multi-year reservoirs. Other values for this threshold were also suggested in the literature; e.g., Wu and Chen (2012) used a $c$ value of 0.3. In this study, scenario A is used for reservoirs that have multi-year capacity ($c>0.5$) and scenario B for reservoirs that have within-a-year capacity ($c<0.5$). Lastly, in Zone 4 (emergency storage zone) the reservoir algorithm operates to avoid reservoir overtopping by releasing the larger of

the maximum release target or all excess storage above the maximum storage value (the flood storage) constrained to the downstream channel capacity $Q_{mc}$. If not specified, a rough estimate of the downstream channel capacity value could be the 99 percentile of non-exceedance probabilities of discharges from historical data.

| *Zone 0* | $Q_t = 0$ | $[S_t < 0.1 S_{max}]$ | (2) |
|---|---|---|---|
| *Zone 1* | $Q_t = min\left(Q_{ci}, \frac{S_t - 0.1 S_{max}}{\Delta t}\right)$ | $[0.1 S_{max} < S_t \leq S_{ci}]$ | (3) |
| *Zone 2* | $Q_t = Q_{ci} + (Q_{ni} - Q_{ci})\frac{(S_t - S_{ci})}{(S_{ni} - S_{ci})}$ | $[S_{ci} < S_t \leq S_{ni}]$ | (4) |
| *Zone 3A* | $Q_t = Q_{ni} + (Q_{mi} - Q_{ni})\frac{(S_t - S_{ni})}{(S_{mi} - S_{ni})}$ | $[S_{ni} < S_t \leq S_{mi}]$ | (5A) |
| *Zone 3B* | $Q_t = Q_{ni} + max\{(I_t - Q_{ni}), (Q_{mi} - Q_{ni})\}\frac{(S_t - S_{ni})}{(S_{mi} - S_{ni})}$ | $[S_{ni} < S_t \leq S_{mi}]$ | (5B) |
| *Zone 4* | $Q_t = min([max\left(\frac{(S_t - S_{mi})}{\Delta t}, Q_{mi}\right)], Q_{mc})$ | $[S_{mi} < S_t]$ | (6) |

where $I_t$, $Q_t$ and $S_t$ are inflow, release and storage at time step $t$. $S_{ci}$ , $S_{ni}$ and $S_{mi}$ are critical, normal and maximum storage targets for month $i$. $Q_{ci}$ , $Q_{ni}$ and $Q_{mi}$ are critical, normal and maximum release targets for month $i$. $Q_{mc}$ is maximum channel capacity parameter.

### 3.2 Evaluation Criteria

       We evaluated the performance of the proposed reservoir operation model in emulating the outflow and storage data

collected for many reservoirs around the world. As this model was intended to be integrated into large-scale H-LSMs, we further evaluated it when embedded in the MESH model (Modélisation Environmentale–Surface et Hydrologie) (Pietroniro et al., 2007). For all of these evaluations, we used Nash-Sutcliffe Efficiency (NSE) (Nash and Sutcliffe, 1970) and Kling-Gupta

Efficiency (KGE) (Gupta et al., 2009) as the metrics to assess the goodness of fit of the model to observed reservoir outflow and storage data.

### 3.3 Identification of Reservoir Operation Model Parameters

As demonstrated in Section 3.1, the proposed reservoir operation model has six parameters ($S_{ci}, S_{ni}, S_{mi}, Q_{ci}, Q_{ni}$

and $Q_{mi}$) that can vary for different times of the year. We recommend varying these parameters on a monthly basis, while other time resolutions are also possible. To normalize the parameters and their ranges across different types and sizes of reservoirs, for every reservoir, we use cumulative distribution functions (CDFs) of historical storage and release values; see Fig. 2 for example CDFs of the Lake Diefenbaker reservoir (Gardiner dam) in the Saskatchewan River Basin, Canada. Our preliminary analysis indicated that target storage and release values corresponding to 10%, 45%, and 85% non-exceedance probabilities

generally perform reasonably well. We call these our 'generalized parameterization'.

However, optimal values of parameters for a given reservoir can be identified, when data are available, through optimization and parameter identification techniques (Maier et al., 2019; Guillaume et al., 2019). For this purpose, we used a bi-objective optimization approach, as follows, that begins with the generalized parameter values as the starting point and optimizes the model fit to both storage and release data simultaneously:

$$\underset{x \in \Omega}{\text{maximize}} \qquad F(x) = (f_1(x), f_2(x)) \qquad\qquad (9)$$

where x is a vector of decision variables (parameter values), $\Omega$ is decision space, $f_1(x)$ is NSE(flow) measuring the goodness-of-fit in reproducing observed release, and $f_2(x)$ is NSE(Storage) measuring the goodness-of-fit in reproducing observed storage dynamics.

For parameter identification on a monthly basis, a total of 72 decision variables were used in the optimization. We chose rather

arbitrarily the storage and release target intervals that correspond to [5-35%], [35-75%], [75-95%] non-exceedance probabilities as the ranges of variation for critical, normal, and maximum (flood) storage and release, respectively.

The bi-objective optimization problem to calibrate 72 reservoir target release and storage parameters was conducted using the AMALGAM evolutionary multi-objective optimization algorithm (Vrugt and Robinson, 2007). AMALGAM was

selected because it provides effective and reliable solutions for multi-objective optimization using multiple search operators (genetic algorithm, particle swarm optimization, adaptive metropolis search, and differential evolution) and self-adaptive offspring creation. Vrugt et al. (2009), Wöhling and Vrugt (2011); Zhang. (2011); Raad et al. (2009); Dane et al. (2010) and others showed that the performance of AMALGAM model parameter calibration was better than or equivalent to some other calibration algorithms across different complex response surfaces. AMALGAM was run using an initial population size of

100, resulting in a total of 15,000 model evaluations to estimate final Pareto solutions for every single reservoir.

## 3.4 Comparison of reservoir operation models

We compared the performance of our DZTR model against those of Hanasaki et al. (2006) and Wisser et al. (2010) using NSE and KGE performance metrics defined on both storage and release simulations. The comparisons were made only for selected non-irrigation reservoirs because their irrigation reservoir formulation requires additional data on water demands. For the method of Wisser et al. (2010), reservoir release was estimated under two conditions as shown in Equation 10.

$$Q_t = \begin{cases} \kappa I_t & I_t \geq I_m \\ \lambda I_t + (I_m - I_t) & I_t < I_m \end{cases} \tag{10}$$

where $\kappa$ and $\lambda$ are empirical constants set to 0.16 and 0.6 respectively and $I_m$ is the mean annual inflow (m³/s) and $I_t$ is inflow to the reservoir (m³/s) at time t.

In the method of Hanasaki et al. (2006), the release from non-irrigation reservoirs was estimated by multiplying the mean annual inflow by release constraining coefficients (Equation 11). The release constraining coefficients for every given operational year were estimated by dividing the initial storage of that year by the maximum storage (equation 12). The start of the operational year was considered to be the month when the mean monthly inflow shifts from being greater to being lower than the mean annual inflow.

$$r_{m,y} = \begin{cases} k_{rls,y} * r'_{m,y} & (c \geq 0.5) \\ \left(\frac{c}{0.5}\right)^2 * k_{rls,y} * r'_{m,y} + \left(1 - \left(\frac{c}{0.5}\right)^2\right) * i_{m,y} & (0 \leq c < 0.5) \end{cases} \tag{12}$$

$$k_{rls,y} = \frac{S_{first,y}}{\alpha * S_{max}} \tag{13}$$

where $c$ is the ratio of maximum reservoir storage to the mean total annual inflow; and $k_{rls,y}$ is the release coefficient; $r'_{m,y}$ is the provisional monthly release (m³/s) which is equal to mean annual inflow (m³/s); $\alpha$ is a dimensionless constant set to 0.85. Equation 12 differentiates between multi-year and single year storage reservoirs based on a threshold value of 0.5 for c.

## 3.5 MESH Modelling System

MESH is Environment and Climate Change Canada's Land Surface-Hydrology Modelling System (Pietroniro et al., 2007) and has been widely used in different parts of Canada (Davison et al., 2016; Haghnegahdar et al., 2017; Yassin et al., 2017; Sapriza-Azuri et al., 2018; Berry et al., 2017). MESH is a grid-based modelling system composed of three components: (1) the Canadian Land Surface Scheme (CLASS) (Verseghy, 1991; Verseghy et al., 1993), (2) lateral movement of surface (overland) runoff and sub-surface water (interflow) to the channel system within a grid cell and (3) hydrological routing using WATROUTE from the WATFLOOD hydrological model (Kouwen et al., 1993).

Currently, the reservoir representation in MESH model is rudimentary. MESH offers two approaches to account for reservoir operation. In the first approach, the observed reservoir release rate at the reservoir location is provided as input to the model. In this approach, the flow from the catchment upstream of the reservoir is discarded as the release is replaced by

observations, a process referred to as "*streamflow insertion*", which limits the utility of the model to simulate future scenarios for which releases are not yet known. This approach violates the water conservation law in the model and also creates discontinuities within the model setup, especially if there are reservoir cascades. Nevertheless, streamflow insertion could be used when coupling water management models with MESH, and these coupled models could be used to formulate scenarios for reservoir operations. As mentioned in the objectives, however, model coupling is not the focus of this study as we are looking to examine the internal representation of reservoir operations within CMs and LSMs. The second approach is a natural lake or uncontrolled reservoir representation model similar to that of Döll et al., (2003), which was shown to be unsuitable for highly managed reservoirs. To improve the reservoir representation in MESH, this study aims to incorporate the DZTR model for controlled reservoirs into the MESH framework and evaluate its performance.

## 3.6 Case Studies and Data

The data set required to build and evaluate a reservoir operation model includes (1) reservoir physical characteristics such as the volume-level-area relationship and maximum capacity, which are static (in the absence of sedimentation or dam heightening), (2) time series of hydrologic variables such as inflow, release, and water level (or storage), and (3) environmental flows. In this study, we assembled such a dataset for 37 reservoirs located in several regions across the globe (Fig. 3) to test the model. These dams represent a wide range of storage sizes, from $0.132 \times 10^9$ m$^3$ to $162 \times 10^9$ m$^3$, spanning multiple orders of magnitudes. Most of these are located in the Western US and Western Canada, while some are located in Vietnam, central Asian countries and Egypt. Table 1 provides a summary of reservoir locations, construction years, main purposes, data periods, and other dam characteristics. Measured inflow, release, and storage time series were collected from different sources. For reservoirs located in Canada, the data were acquired from Water Survey Canada, Alberta Environment and Parks, and the Saskatchewan Water Security Agency. Data for the High Aswan dam were acquired from the Nile Basin Encyclopaedia via the Nile Basin Initiative. The data for other reservoirs were provided by the authors of previous studies (Hanasaki et al., 2006 and Coerver et al., 2018). Additional information about the degree of regulation, dam height, and catchment area were obtained from GRanD database (Lehner et al., 2011). Reservoir operation simulations were performed on daily and monthly bases with simulation periods varying from 8 to 62 years. The choice of simulation period and time scale was based on data availability (Table 1). The first year of the reservoir simulations was used for spin-up, while the first half of the remaining data periods were used for calibration and the second half for model validation.

We also evaluated the integration of our reservoir model into the MESH model on six reservoirs in two major basins in Western Canada. Six of the test reservoirs (Gardiner, St Mary, Waterton, Oldman, Ghost and Dickson dams) are located within the heavily regulated Saskatchewan River basin (SaskRB) an done reservoir (Bennet dam) is located in the Mackenzie River Basin (MRB). For both of the basins, the MESH model was set up on a grid resolution of 0.125° and the data required to build the MESH model were obtained from different sources. The topographic data are based on the Canadian Digital Elevation Data (CDED) at a scale of 1:250,000 and were obtained from the GeoBase website (http://www.geobase.ca/). The data on seven climate forcing variables at a 30-min temporal resolution were obtained from Global Environmental Multi-scale

(GEM) NWP model (Côté et al., 1998) and Canadian Precipitation Analysis (CaPA) (Mahfouf, et al., 2007). The land cover data used are based on 2005 land-cover map from the Canada Centre for Remote Sensing (CCRS). Soil texture data were obtained from Soil Landscapes of Canada (SLC) data of Agriculture and Agri-Food Canada. The MESH parameter values were taken from previous studies for calibration to streamflow at major subbasins of SaskRB and MRB.

**4 Results and Discussion**

**4.1 Evaluation of the dynamically zoned target release (DZTR) model with generalized parameters**

Individual reservoir simulations were conducted by the DZTR model with generalized monthly storage and release parameter values set at non-exceedance probabilities recommended in Section 3.3 for representing the reservoir storage zones and their respective target releases. The evaluation of the DZTR model was based on the performance metrics and a comparison with the other reservoir operation approaches and a base case where the existence of a reservoir was ignored in a model,
referred to as the "no-reservoir assumption". Under the no-reservoir assumption, the release was considered equal to inflow, and storage was considered constant, and as such, the performance metrics were computed by directly comparing inflow with observed release.

Fig. 4 shows performance metrics results of the DZTR model in terms of NSE and KGE for storage and release
simulations compared to those of the base case. As shown in Fig. 4a, both NSE (Flow) and NSE (Storage) results are greater than 0.25 and 0.5 for 90% and 50% of reservoirs, respectively. Although NSE (Flow) results are greater than zero for all reservoirs, 1% of reservoirs resulted in a negative NSE (Storage) values. The no-reservoir assumption resulted in NSE (base-case) values of greater than 0.25 and 0.5 for 45% and 30% of reservoirs respectively, which, in general, are much lower than those of the DZTR model. Under the no-reservoir assumption, 48% of the reservoirs resulted in a negative NSE (base-case).
Almost all positive NSE (base-case) results were observed on reservoirs with $c<0.5$ such as Dickson, E.B. Campbell, Kayrakkum, Oldman and Tyuyamuyun (as explained in Section 3, $c$ is the ratio of storage capacity to annual inflow volume). However, for reservoirs with $c>0.5$ such as Bhumibol, Flaming Gorge, Fort Peck, High Aswan, W.A.C. Bennett, the NSE (base-case) is negative, which indicates the significant influence of their regulations on the hydrograph shape. Similarly, Fig. 4b shows the evaluation of the different reservoir models based on the KGE metric (Gupta et al., (2009). The values of KGE
(Flow) and KGE (Storage) are greater than 0.25 and 0.5 for 100% and 86% of the reservoirs, respectively. The KGE (base-case) values of 21% of reservoirs are less than 0, while those of 57% and 49% of the reservoirs are greater than 0.25 and 0.5, respectively. The NSE and KGE results show that the DZTR with the generalized parameter values is capable of simulating flow and storage simulation well.

Fig. 5 shows scatter plots between KGE, NSE, and the regulation level represented by $c$. These plots orientation of
the scatter plot between NSE and KGE on flow and storage show a strong positive correlation between the evaluation metrics which indicates that both metrics provide somewhat similar evaluation information. Fig. 5a and 5b show that both no-reservoir assumption and DZTR estimate the release more accurately for lower levels of regulation. As expected, the degradation of

performance was pronounced for no-reservoir assumption as the regulation level increased, while DZTR performance reduced by a much smaller extent (still positive values). Almost all low regulation level reservoirs ($c<0.5$) showed positive performance metrics which means the reservoir regulation does not strongly modify the flow regime, whereas the opposite case is true for highly regulated reservoirs ($c>0.5$) in which the reservoir regulation strongly changes the reservoir release. Coerver et al.

(2018) also noted that low regulation level reservoirs are more dependent on the current time step inflow knowledge because their smaller influence on the flow regime. The method of Hanasaki et al. (2006) also recognizes the strong dependence of $c<0.5$ reservoirs on inflow to determine the release by configuring the release as a function of monthly mean inflow. Conversely, the relationship between the regulation level and the storage simulation performance (in terms of both KGE (Storage) and NSE (Storage)) did not show a strong correlation (Fig. 5c).

Fig. 6 compares the reservoir simulation and observation time series for the whole simulation period, while Fig. 7 shows the long-term average of these simulations. Inflows are also included in Fig. 6 and Fig. 7 to show the regulation pattern and changes caused by reservoir operation. Both figures indicate that the DZTR model captures both release and storage dynamics well, reproducing the daily and monthly seasonality as well as the magnitude and timing of storage and releases for almost all reservoirs, especially for reservoirs with high regulation (multipurpose, multiyear reservoirs) such as American

Falls, Bhumibol, High Aswan, Sirikit, Trinity, and Bennett dams. However, the simulations also show some systematic over- and under-estimations; for example, the simulations of Bhumibol, Fort Peck, High Aswan, Int. Falcon, Navajo, Bennet, and Int. Amistad reservoirs show continuous underestimation and overestimation of reservoir storage. Some reservoirs such as Trinity, Palisades, Kayrakkum, Flaming Gorge, and Garrison show underestimation and overestimation of reservoir storage only for some seasons. A closer look at American Falls, Flaming Gorge, Fort Peck, Glen Canyon, Navajo dams in Fig. 6

indicates that the DZTR model reliably captured storage and release seasonality, inter-annual trends, and release pattern shifts during consecutive wet years 1982-1986 followed by consecutive dry years 1987-1993. Similar patterns can be observed for the Gardiner dam with good simulation during both dry years (1984-1986, 1988-1989, 1999-2004) and wet years (1993, 2005, 2010-2011). Furthermore, as expected, Fig. 7 shows that lowly-regulated reservoirs ($c <0.5$) have less impact on the flow regime, but with fairly significant storage seasonality (Oldman, E.B. Campbell, Palisades, Andijan). In general, the DZTR

model with the generalized parameterization of reservoir zones and releases showed an improved performance and can be applied to any hydrological model (CM or H-LSM) that involves reservoir simulation.

It is important to note that for the case of a cascade of reservoirs, the parametrization of the DZTR model implicitly accounts, to some extent, for the upstream regulation effects by the upstream cascade reservoirs. This is because the regulated inflow is used for parametrizing downstream reservoirs, which reflects the regulation information of upstream reservoirs in

the cascade. In reality, the operation of some cascade reservoirs are highly interlinked, particularly during the flood season. The decision regarding the release from one reservoir accounts for the (forecasted) state of other reservoirs. Such dual- or multi-linked operation is however not accurately accounted for in the presented algorithm, because it assumes that each reservoir operates using its own storage state, inflow and target storage and releases. Such systems require detailed modelling

of operations that is not usually attainable in large scale hydrological models. Depending on the purpose of the model, the modeller may decide to lump those reservoirs together to improve simulations downstream as in e.g., Ehsani et al. (2016).

## 4.2 Comparison with previously developed reservoir operation models

To further illustrate the reliability of DZTR model in representing reservoir simulation, a comparison with the methods of Hanasaki et al. (2006) and of Wisser et al. (2010) was conducted as shown in Fig. 8. The comparison shows that the DZTR model provides a considerable improvement according to all of the performance criteria, notably NSE (Storage) and NSE (Flow), except in the case of the E.B. Campbell dam where Hanasaki et al.'s method showed similar performance to DZTR. Also, the method of Hanasaki et al. (2006) outperformed that of Wisser et al. (2010). Out of the thirteen reservoirs compared, the DZTR resulted in positive values for both NSE (Storage) and NSE (Flow) for all except for E.B. Campbell storage. The method of Hanasaki et al. (2006) and Wisser et al. (2010) resulted in eight and five reservoirs with positive NSE (Flow) respectively (Fig. 8a), while both produced negative values for NSE (Storage) for all the reservoirs compared (Fig. 8c). A similar performance pattern was observed for KGE metrics for flow and storage. In addition, we compared the DZTR result shown in Fig. 7 and 8 with the results reported in Coerver et al. (2018) who applied a fuzzy-neural network model to extract 11 operating rules. This comparison showed that the performance of our generalized parameterization is comparable to that Coerver et al. (2018) in simulating reservoir release; note that performance on storage is not reported in Coerver et al. (2018). This indicates that the simple parameterization applied in the DZTR model can provide a solution that is at least as effective as that of a neural network-based model. Equally importantly, the DZTR model is transparent, as opposed to neural network methods that are often criticized as being a "black box".

The above comparisons were conducted for non-irrigation reservoirs because water demand data is needed to use the Hanasaki et al. (2006) method for irrigation reservoirs. In the case of the DZTR approach, the idea is that the DZTR model operates in such a way to infer existing operational rules which cater for those demands. Thus, the release from DZTR accounts, implicitly, for downstream demands as per the intended purpose of the reservoir whether it is for flood control, irrigation, hydropower, etc. or any combination of these. The case study dams in our study include reservoirs with different purposes as shown in Table 1. The DZTR approach showed good performance for reservoirs with different purposes.

If the reservoir purpose is irrigation, the target releases from DZTR are to satisfy irrigation demands because the parameterization is optimized based on observed releases. The release from an irrigation dam will be available for abstraction at the predefined abstraction points downstream of the dam. The abstraction and distribution can be implemented as separate modules as done within the MESH land surface model (Yassin et al., 2019). In such an implementation, MESH takes care of (1) calculation of actual irrigation demand for a configured irrigation area, (2) water abstraction from defined abstraction point along the river below the dam and (3) distribution across the irrigation fields. Regarding the return flow, the excess water flows from the irrigation areas are assumed to join the nearest stream within the model grid cell.

The DZTR model can in principle handle multi-purpose reservoirs, e.g., a reservoir that is used simultaneously for hydropower generation, irrigation water supply, and flood control (e.g., High Aswan Dam in Egypt which is one of the studied

reservoirs), the DZTR provides the release based on the inflow, and storage conditions and that will be available for irrigation downstream. Hydropower does not consume water but returns it back to the river (except in rare cases where it returns to a different channel). Flood control is directly accounted for in the scheme and becomes relevant when storage is within the flood storage zone. Further, the flexible formulation of DZTR allows to implicitly change the priorities in operation for selected time periods (e.g., months or seasons) by changing the target storage values during flood periods (e.g., the storage target before the onset of snowmelt). During these flood months, lowering the target storage would increase the buffer for flood control. Conversely increasing the target storage during other months would be desirable to store water and release during irrigation months. When the scheme is optimized using inflow, release, and storage data, the parameterizations capture these priorities implicitly as expressed in the data. When inflow data are lacking, the generalized parametrization will set the storage zones based on the suggested exceedance probabilities (that were deduced based on all reservoirs used in the study) and the priorities can be assumed as pre-defined.

## 4.3 Initial storage and inflow sensitivity test

The initial storage at the beginning of the simulation is an input that needs to be specified to the model. The initial values can be prescribed from the observations if available. However, the simulation of a hydrological/land surface model could start at any point in time when there is no observation to prescribe (e.g., some time in far past, a future scenario simulation, or a hypothetical scenario). Additionally, in a long-term simulation, the initial storage may result from a previous model simulation, which may not be as close to observations as desired. The aim of the experiment is to examine and show to what extent the initial storage value affects the simulation performance.

To test the effect of initial storage used in the reservoir simulation performance, two experiments were conducted on three reservoirs with different scale of regulations 1) Charvak (c=0.28), 2) Gardiner (c=1.46), and 3) High Aswan (c=2.84). In the first experiment, the initial storage was allowed to vary between ten percent of maximum storage $(0.1 * S_{max})$ to maximum storage ($S_{max}$). In the second experiment, the initial storage range was narrowed to starting simulation month minimum and maximum historical observations. In both tests, 150 simulations were conducted by sampling the initial storage using uniform random sampling from the defined storage range.

Fig. 9 and Table 2 show the results of these initial storage perturbation experiments. For both experiments the simulations on the Charvak dam showed a similar range for NSE (Flow) [0.79, 0.83] and NSE (storage) [0.61, 0.74]. Using one year as a spin-up period on Charvak dam simulations stabilized the initial storage effects, resulting in NSE (Flow) of 0.82 and NSE (Storage) of 0.74. The simulations on Gardiner dam in the first experiment showed a range of [0.35, 0.51] for NSE (Flow) and [-0.43, 0.88] range for NSE (storage), while in the second experiment the ranges were narrowed to [0.44, 0.49] for NSE (Flow) and [0.87, 0.88] for NSE (storage). For a one year spin-up period on the Gardiner dam this simulation converged the NSE (Flow) range to [0.49, 0.51] and the NSE (Storage) range to [0.76, 0.87] in the first experiment and to 0.49 NSE (Flow) and 0.87 NSE (Storage) for the second experiments. On the other hand, the simulation on the High Aswan dam showed a range of [-0.28, 0.85] for NSE (Flow) and [0.38 0.91] for NSE (storage) for the first experiment and [0.52, 0.85] for NSE

(Flow) and [0.42 0.91] for NSE (storage) for the second experiment. Excluding a one year spin-up period from the metric calculation on the High Aswan dam simulation narrowed the NSE (Flow) range to [0.62, 0.85] and the NSE (Storage) range to [0.58 0.91] for both experiments. Overall, as expected, the experiments suggest that the effect of initial storage on reservoir simulation performance depends on the regulation scale. Starting from observed storage values and using a one-year warm-up period allows stabiliztion of the initial storage effect for low and medium regulated reservoirs. However, for highly regulated reservoirs, as in the case of High Aswan, longer spin-up periods are needed to stabilize the simulations. For example, a five-year spin-up period was required to fully stabilize the performance for the High Aswan dam simulations.

The existence of inflow bias is inevitable in any hydrological modeling practice. To understand the behaviour of the DZTR model under biased inflow conditions, we conducted a sensitivity experiment on the Charvak, Gardiner and High Aswan reservoirs. To do so, the DZTR model performance was tested using five simulations in which the entire inflow time series was changed by -50%, -25%, 0%, +25%, and +50%. The sensitivity of simulations to bias in inflow was evaluated using the NSE (Flow) and NSE (Storage) performance metrics.

Fig. 10 and Table 3 show the results of the inflow bias test and that the reservoir simulation performance significantly changes as a result of this bias. Reducing the inflow by 50% considerably reduced the reservoir storage and release and led to negative values of NSE (Flow) and NSE (Storage) for all reservoirs. For such a large negative inflow bias, the reservoir operation tries to recover the storage to the target (observed) level by releasing a low as possible. Conversely, the positive inflow bias increased simulated storage and releases for all reservoirs, which led to negative performance metrics for all reservoirs except on Gardiner NSE (Storage). As shown in Fig. 10, with large positive inflow bias, storage quickly moves towards flood and maximum storage targets resulting in insufficient storage left to attenuate flood peaks and the operation model starts discharging large releases through the spillway to maintain the storage at the maximum storage target. Inflow bias of -25% and +25% showed similar behaviour as -50% and +50% bias for all reservoirs, but the simulation performance metrics during -25% and +25% provide significant positive NSE values for the Charvak and Gardener dams except for the Gardiner NSE(Flow) for +25% which resulted a negative NSE value. However, on the highly regulated High Aswan dam, the ±25% inflow bias significantly reduced the performance to negative values.

## 4.4 Parameter calibration and validation of the DZTR model

We tried to improve upon the generalized parameterization by calibrating the DZTR parameters via bi-objective optimization for two objective functions, Nash Sutcliffe on reservoir storage (NSE (Storage)) and Nash Sutcliffe on reservoir release (NSE (Flow)). This is an important step when the data and computational resources for optimization are available, to enhance reservoir simulation and consequently hydrological modeling of the region of interest. Fig. 11 shows the multi-criteria reservoir calibration (yellow circles) and validation (red circles) Pareto solutions for all reservoirs. The Pareto solutions show strong tradeoffs between fitting observed reservoir storage versus downstream release, which also reflects the fact that the problem is multi-objective by nature and it is required to consider both storage and release, instead of fitting one at the cost of degrading the other. The generalized parameterization solution for the calibration (Yellow Square with blue border) and

validation periods (red square with blue border) is also added in Fig. 11 for each reservoir to show the improvement gained through parameter calibration. Relative to the generalized solution for the calibration period, reservoir parameter calibration improved both NSE (Flow) and NSE (Storage) for all reservoirs with a median improvement of 0.11 and 0.21, respectively. The NSE (Flow) improvement ranged from 0.017 to 0.575, and NSE (Storage) improvement ranged from 0.02 to 0.66. The

Parameter calibration has shown significant improvement on reservoirs that have lower performance with generalized parameterization. The best examples of this case are Fort Randall, Int. Amistad, Trinity, Int. Falcon, and E.B. Campbell, as shown in Fig. 11. Small improvements in performance have also been observed on reservoirs that have greater performance with generalized parameterization such as American Falls, Andijan, Nurek, High Aswan, Waterton, and Charvak. The validation of calibrated solutions improved the NSE (Flow) and NSE (Storage) for 56% of the reservoirs with a median

improvement of 0.035 and 0.092, respectively. The NSE (Flow) improvement in the validation period ranged from 0.001 to 0.335, and NSE (Storage) improvement ranged from 0.004 to 1.02. During validation, the remaining reservoirs (44% of them) resulted in NSE (Flow) and NSE (Storage) reductions with a median reduction of 0.032 and 0.089, respectively. The reductions of NSE (Flow) ranged from 0.001 to 0.073, and those of NSE (Storage) ranged from 0.001 to 0.257.

Overall, considerable improvement was achieved for both calibration and validation periods for several reservoirs

such as the Dickson, Gardiner, Ghost, Int. Amistad, Int. Falcon, Kayrakkum, Sirikit, Yellowtail, and Glenmore. However, as shown in Fig 11, the improvements of DZTR model performance during calibration do not usually guarantee performance improvement in validation. This is because, as for any other types of model as well, the properties of the calibration and validation periods might differ significantly. In particular, the calibrated Pareto solution does not show the same trade-off or level of performance during validation when there is considerable change in inflow properties as a result of consecutive wet

or dry years. Examples of this condition are shown for Glen Canyon (similarly Bhumibol, Fort Randall, and Fort Peck) where the calibration period had more wet and high inflow years than the validation period. Such considerable changes of inflow, storage, and release results in performance degradation during the validation period. In general, a small change in inflow, storage, or release for the validation period can change the shape of the trade-off. However, the calibrated parameters in most cases were still capable of producing good performance during validation close to or better than that of the generalized

parameterization for the same period.

To further test the role of the calibration period, we calibrated all reservoirs using the whole observational record. The result of this test is shown in Fig. 12 which demonstrates the strong role of the calibration period. All reservoirs showed trade-off between storage and release fitting. The solution resulted in a consistent Pareto pattern similar to the split-sample calibration results. The median NSE (Flow) and NSE (Storage) improvement when using the whole observational record for

calibration are approximately 0.1 and 0.12 respectively, while the maximum improvement reached 0.45 and 0.55 for some reservoirs. High improvements on storage and flow simulations in the case of whole-period-calibration are mostly observed on reservoirs that have considerable shift of observed storage and flow across the period of observation period. Fig. 13 shows some example reservoirs that had considerable improvements such as Bhumibol, Canyon Ferry, Int. Amistad, Int. Falcon,

Navajo, and Trinity dams, compared to generalized parameters (Fig. 6). Similarly, for the remaining reservoirs, calibrating the whole period showed (Fig. 13) better agreement of daily and monthly simulations with the observations, even for years with extreme deviations that are most likely associated with extreme dry and wet conditions. Additionally, the long-term average simulations (Fig. 14) showed that calibrating using the whole period reduced the deviation between simulations and observations, and in most cases the Pareto simulation range encompasses the observation. Overall, the calibration period test indicates the benefit of using long-term observation for parametrization (even for generalized parameterization) to allow the parametrization to represent behaviour in extreme periods. Thus, we recommend using as much data as available to parameterize the model for a specific reservoir so that all information on reservoir operation will be accounted for.

The DZTR scheme introduces more parameters to the host land surface model. However, its parameters are external to those of land surface model and are determined a priori using storage and release data. The decision of the time scale to use for specifying the parameters is left to the modeller. The user has the ability to investigate the seasonal patterns in the storage and release data and decide whether a monthly or a coarser time scale (e.g., quarterly) would be sufficient. In fact, the configuration of DZTR is also flexible to use any user-specified zoning that are available from observation, reservoir information or zoning values specified in other studies such as Zhao et al. (2016).

## 4.5 DZTR model test within the MESH model

Finally, the generalized parametrization of the DZTR model was integrated into the MESH model and tested to simulate six reservoirs in the Saskatchewan River Basin (Gardiner, St Mary, Waterton, Oldman, Ghost and Dickson dams) and one reservoir (Bennet dam) in the Mackenzie River Basin, both in Western Canada. The reservoir simulation was run using MESH modelled inflows at a half-hourly time step, the usual MESH time step, and the performance metrics were calculated at a daily time step. The MESH modelled inflows are considered to represent the base-case scenario, and the inflow can be assumed as regulated or natural depending on whether there are dams upstream or not.

Fig. 15 illustrates that the generalized DZTR model generally improves upon having no representation of the reservoirs in the model. This improvement is apparent in the NSE values of the flow, which increase with the DZTR model. The only exception is Dickson dam with a small reduction in NSE. The importance of integration of the DZTR model was predominant for the Gardiner and Bennett dams, which are highly regulated reservoirs (c>0.5) when compared to the other reservoirs tested in MESH.

This general improvement of flow simulation when comparing a reservoir model to the no-reservoir assumption is, of course, not surprising. What is important to note, however, is that the improvement in NSE can be dramatic without calibration of the DZTR parameters. This is important for many LSM applications where calibration is generally not performed. Hanasaki et al. (2006) illustrated that their method is superior to the natural lake (or unregulated reservoir) method applied in many CMs and H-LSMs, and this paper shows that the DZTR model improves upon the results of Hanasaki et al. (2006). Therefore, it is natural to assume that the DZTR model would also be an improvement in uncalibrated H-LSM applications.

However, calibration is very common in CM or H-LSM applications in which the DZTR model would likely be employed. A full comparison of calibrated results between a no-reservoir case, natural lake (or unregulated reservoir), and the DZTR model (and the other reservoir models) is beyond the scope of this paper. Again, given the improvements shown with the uncalibrated DZTR model when compared with other uncalibrated models, and the general improvements shown here when calibrating the DZTR model, it is assumed that calibrating the DZTR model within a CM or H-LSM would improve upon calibrating an unregulated reservoir model, or the other reservoir models compared in this paper.

The storage simulation showed low NSE (Storage) value for St. Mary and Waterton dams and negative NSE (Storage) for Oldman and Ghost dams. However, the simulation showed a reasonable representation of storage variability, but with considerable underestimation. This underestimation in storage in Fig. 15 is attributable to the fact that the modelled inflow is underestimated. It is expected that calibration of the land-surface parameters in conjunction with the DZTR parameters in MESH would improve the modelled inflows and resulting modelled reservoir storage.

It is worth mentioning again that H-LSMs, such as MESH, can also be used for the original purpose of LSMs, which is to represent fluxes from the land-surface to the atmosphere. If the approach improves modelled flows where reservoirs operate, it could result in a better parameterization of the LSM, which should in-turn improve land-surface fluxes and feedbacks to the atmosphere.

### 4.6 Uncertainties in reservoir operation and DZTR parameterization

Reservoir operation on its own involves considerable uncertainties that is attributed to several factors. One major source of uncertainty in reservoir operation is future inflows (long-term and short-term inflow forecast). The forecast contains errors rooted in the forecast method, the driving climate forecast, snowpack measurements, timing of snowmelt and the statistical (stationarity) assumptions to generate inflows based on historical inflows. The inflow forecast uncertainty is more significant during flood seasons because it involves subjective decisions of operators to avoid the risk of dam overtopping and downstream flooding. Other sources of uncertainty in reservoir operation include changes in demand over time because of increases in demand for irrigation, power, water supply, etc. The purpose of the reservoir can also change from its initial intended purpose (e.g., adding a hydropower station to an irrigation dam). These changes are only implicitly captured by the DZTR scheme as implied in the storage and release time series used for parameterizing it for a specific reservoir.

Given the above uncertainties, even the actual reservoir operation may deviate from the designed reservoir operation rule curve. Some of the decisions of reservoir operators are spontaneous, ad-hoc, and depend on experiences that are not usually documented. Thus, there are difficulties to accurately represent the historical operation or to establish accurate relationships between reservoir storage, inflow, and release. These relationships typically contain considerable noise e.g., different release values for the same storage level during the same season. As a result, these uncertainties considerably influence the parameterization of the model derived to represent the reservoir operation based on historical observations of each reservoir. This is particularly true for the algorithm presented because of two main factors. Firstly, the presented reservoir algorithm assumes that the relationship between reservoir storage and releases follow piecewise linear functions. There is a

chance that other functional forms represent such relationship better for some reservoirs. Secondly, in the case of the generalized parameterization, the bending points in the piece-wise linear functions (zone classification points) are estimated based on fixed probabilities of exceedance extracted from historical data for all reservoirs. A different dataset (of reservoirs and/or time periods) could result in different quantiles. The assumption of having similar bending points of the piecewise linear functions for all reservoirs cannot provide optimal zones for each reservoir. However, we showed that the generalized parameterization performs better compared to other widely used algorithms.

Optimizing storage and release parameters allows to overcome the limitation of generalized bending points of the piecewise linear function by adjusting the bending points so that the best fit can be identified. However, optimization usually does not provide a perfect storage release relationship (i.e., in general, the trade-off between objectives never converges to single point), because the perfect representation only happens in the case of a perfect reservoir model and perfect data. The proposed model, like many other types of models is not an exception because of the uncertainties highlighted in the previous point. Thus, the trade-off between storage and release objectives can be viewed as a measure of the limitation of the reservoir algorithm (piece-wise linear functions, fixed number of zones, etc.) and observation errors. To examine the level of uncertainty of the trade-off, it is important to look at the shape and range of the trade-off on each objective function axis.

As shown in Fig. 11 and Fig. 12, except for few reservoirs, the range of Pareto solutions for each objective function is generally narrow with good NSE values. In such cases, the associated uncertainties are less and the trade-off between improving simulated releases and improving simulated storage is minimal. Conversely, in some cases, an extended spread of the tread-off along one of the axes (objective function) was observed, indicating a higher uncertainty of the algorithm for the process that the axis represent, i.e., reservoir storage or release. This requires further investigations of the datasets and parameterization for those reservoirs and their history of operations. Shifts in operational management of reservoirs do occur and these may obscure the parameterization. These may be detected by careful examination of the available records as well as metadata records of the reservoir history if accessible. The level of noise when determining the parameters could be an indicator of changes in operation.

**4.7 Implementation strategies to overcome data limitation**

The data requirement is the main limitation of the DZTR model for application at continental and global scales. One approach to overcome data limitations is to integrate our proposed method in land surface/catchment models along with other reservoir operation methods (e.g., Hanasaki et al., 2006). Then, within the land surface/catchment models, identifier flags can be used to indicate which method applies to which reservoirs. The DZTR approach can only be activated for reservoirs with data support, while the remaining reservoirs can use other approaches as dictated by data availability. We have been following such an implementation within the MESH model.

As shown in our results, reservoir regulation has a huge impact on downstream flows if the reservoir is highly regulated and/or is of multi-year type ($c>0.5$). Thus, more emphasis can be put on those reservoirs with $c>0.5$. At the moment, such methods will be more effective at the regional than global scale (for example for Saskatchewan River Basin in our case),

because modellers at regional scale have better access to inflow-storage-outflow data and have better understanding of the system to acquire the necessary reservoir data. In a land surface hydrologic model, important reservoirs are those causing large changes to the downstream flows and those tend to be the larger ones with generally better data availability.

Data on reservoir storage, inflow and release exist for most reservoirs but sometimes they are not made publically available. Storage data can be obtained from water level data which is generally available for major reservoirs and can be converted to storage. Release data can be deduced from the nearest downstream station. In addition, new initiatives are needed to gather and archive such reservoir datasets and move beyond information on reservoir characteristics that is currently available in databases (e.g., GRanD database - Lehner et al., 2011). One of our recommendation is that the target release and storage data be archived for public use at least for highly regulated and multi-year type dams ($c$>0.5).

The possibility of estimating storage and release data from different satellite data products is promising; such new data sources will potentially improve the use of methods like the presented reservoir operation (optimized or generalized). More recently, Busker et al. (2019) showed an estimation of volume for 130 reservoirs using surface water dataset and satellite altimetry, which is encouraging.

## 5 Summary and Conclusions

Human interventions in hydrologic systems through dams and reservoirs significantly change the flow regime of many rivers. In this paper, we presented an improved reservoir operation model, called the Dynamically Zoned Target Release (DZTR) model that can be integrated into any large-scale hydrological model; here we integrated it into the MESH hydrology-land surface model. The DZTR model is based on parametric piecewise linear functions that approximate reservoir release rules used by reservoir operators. We proposed two strategies to identify the parameters of this model: one based on the distributions of historical storage and release to generate the so-called "generalized parameters" and the other one based on direct calibration to observed storage and release time series via multi-objective optimization. We first tested the DZTR model individually across a number of reservoirs around the globe, and then tested its performance when plugged into the MESH model for a subset of those reservoirs. Our conclusions can be summarized as:

- The DZTR reservoir operation model performed well in reproducing observed storage and release time series in (almost) all reservoirs tested and outperformed the existing reservoir models proposed by Hanasaki et al. (2006) and Wisser et al. (2010). The model was capable of capturing inter- and intra-annual variability of both reservoir storage and release.

- As expected, calibration significantly improved the performance of the DZTR model compared with the performance of the "generalized parameters". There often exists, however, a significant tradeoff between fitting reservoir storage versus release, signifying the importance of accounting for both storage and release in a multi-objective fashion.

- The integration of the DZTR reservoir model into the MESH hydrology-land surface modelling system was straightforward and improved the overall model performance compared with the traditional methods of accounting

for reservoirs in H-LSMs. This integration can be viewed as a successful example for improving the representation of reservoir operation in CMs, LSMs and GWSMs.

Future research work may include (1) examining the applicability of the DZTR model for regions with severely-limited data by examining the utility of other data sources such those derived from satellite-based observations (Savtchenko et al., 2004; Garambois and Monnier, 2015; Gao et al., 2012) and using area-volume relationship approximated by regular geometric shapes (e.g., Yigzaw et al., 2018); and (2) examining direct one- and/or two-way coupling of WMMs with CMs and LSMs towards developing a seamless coupled framework for the simulation of natural-engineered watershed systems.

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

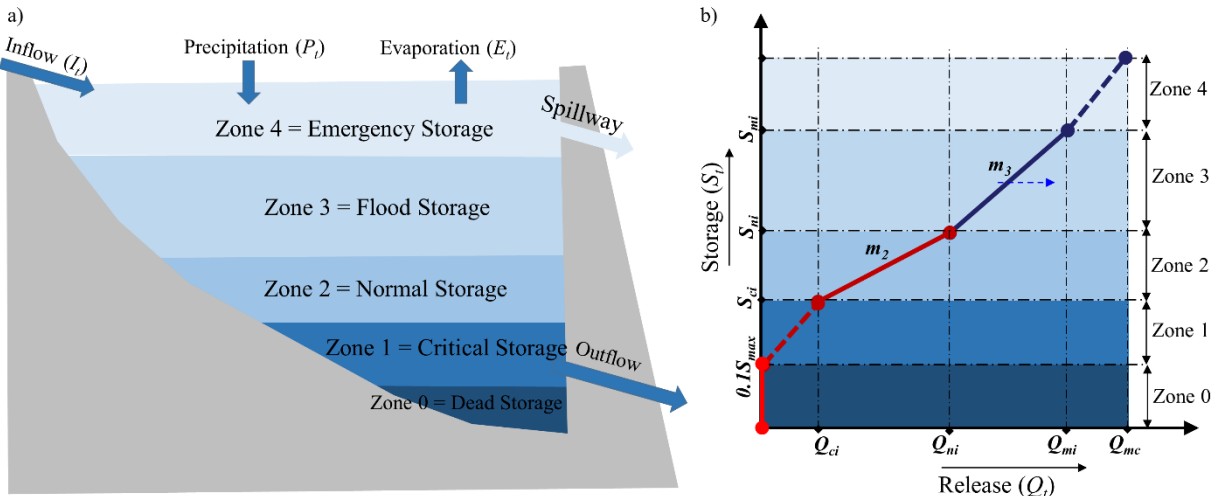

**Figure 1: The schematic representation of reservoir zoning and storage-release function: a) Four (active) reservoir zones with inflow and outflows; b) piecewise linear reservoir release function, $m_1$, and $m_2$ control the slope of the release curve and they change monthly. The upward blue arrow is to indicate that inflow to the reservoir may also be considered in determining the release in zone 3.**

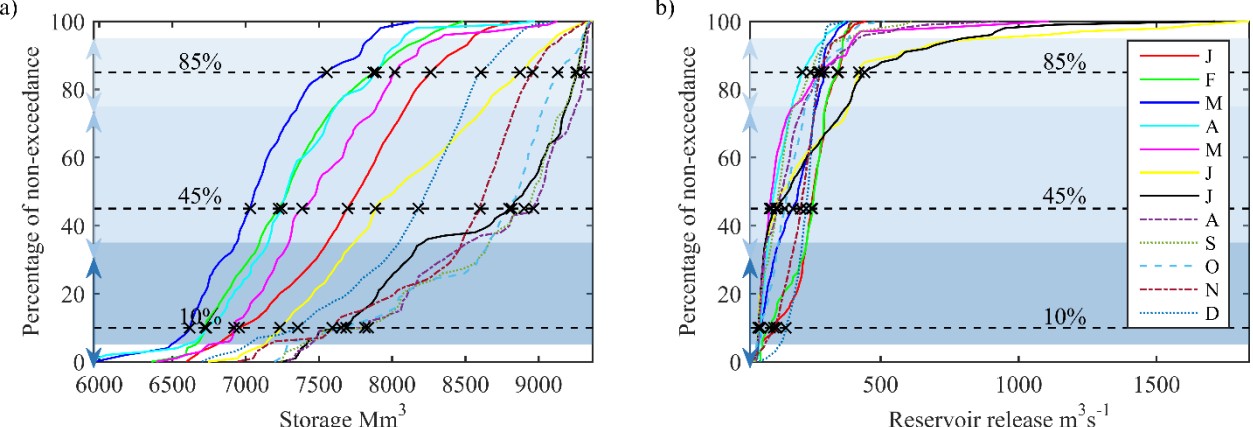

**Figure 2: Cumulative Distribution Function (CDF) a) Storage CDF of Gardiner dam b) Reservoir release CDF of Gardiner dam. Double arrows on y-axis shows parametrizations ranges for each generalized parameters.**

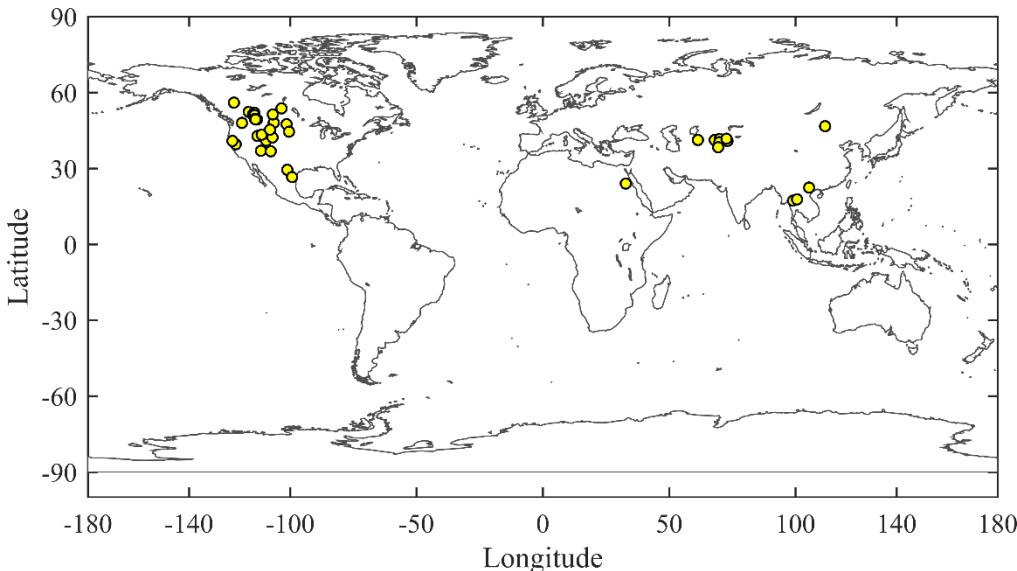

**Figure 3: Locations of dams used to evaluate reservoir routing model.**

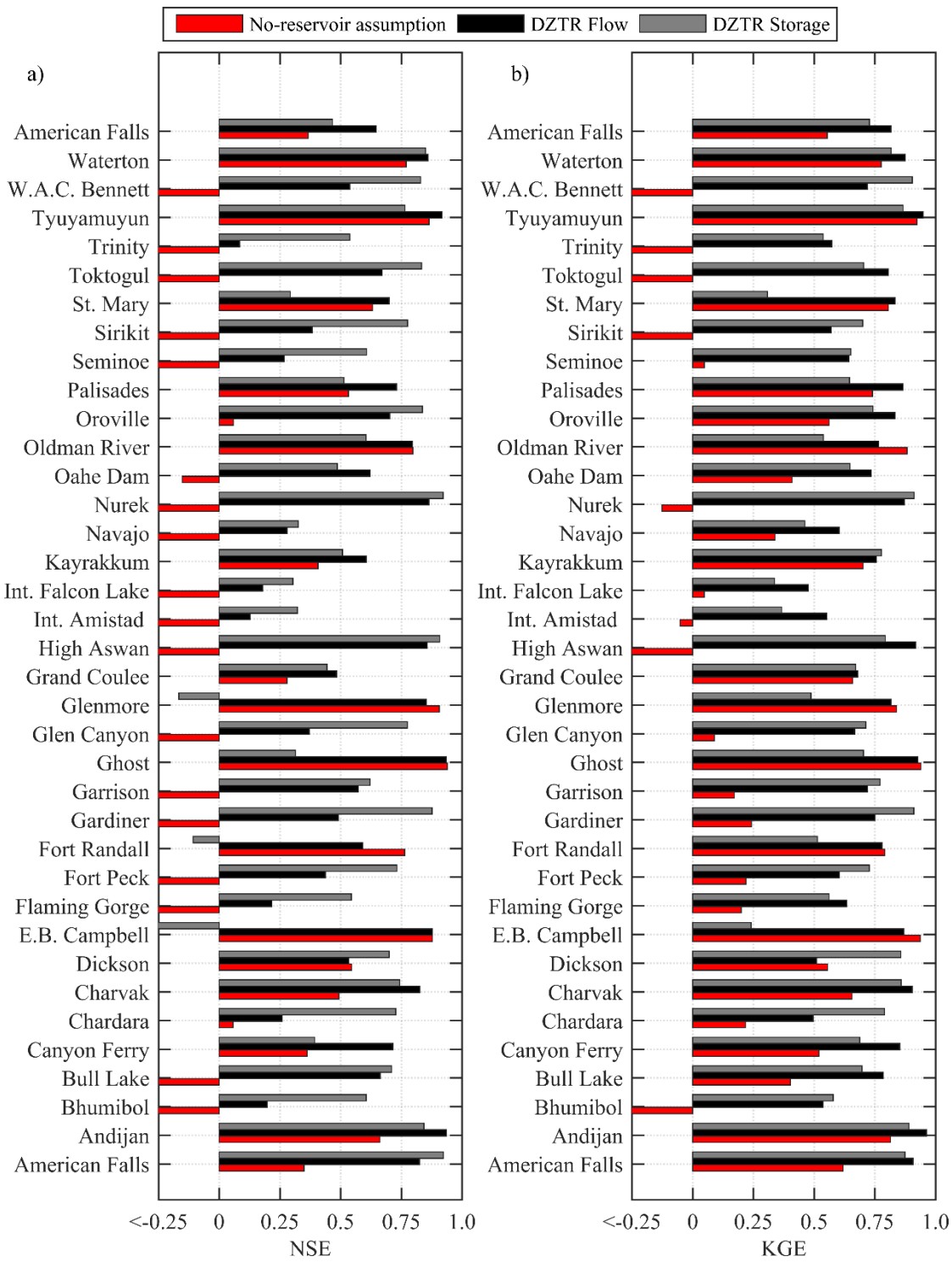

**Figure 4: Performance evaluation result of the DZTR model reservoir operation algorithm a) NSE performance metrics, b) KGE performance metrics.**

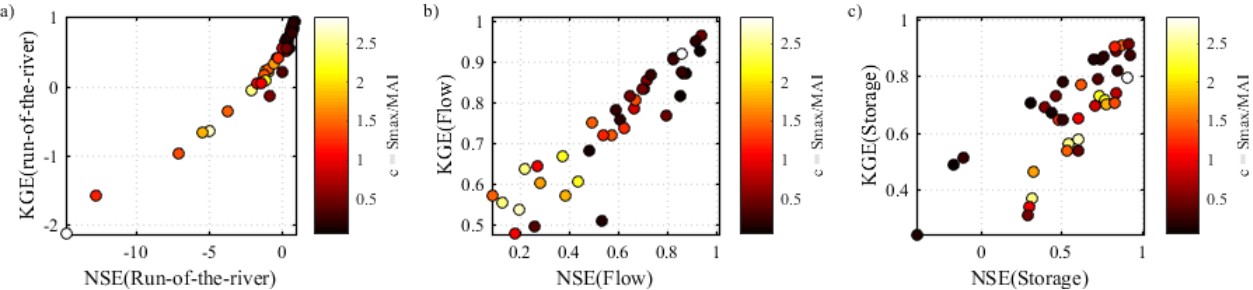

Figure 5: Scatter plot between KGE and NSE with regulation scale represented in terms of c a) KGE and NSE on no reservoir condition, b) KGE and NSE on DZTR release, and c) KGE and NSE on DZTR storage.

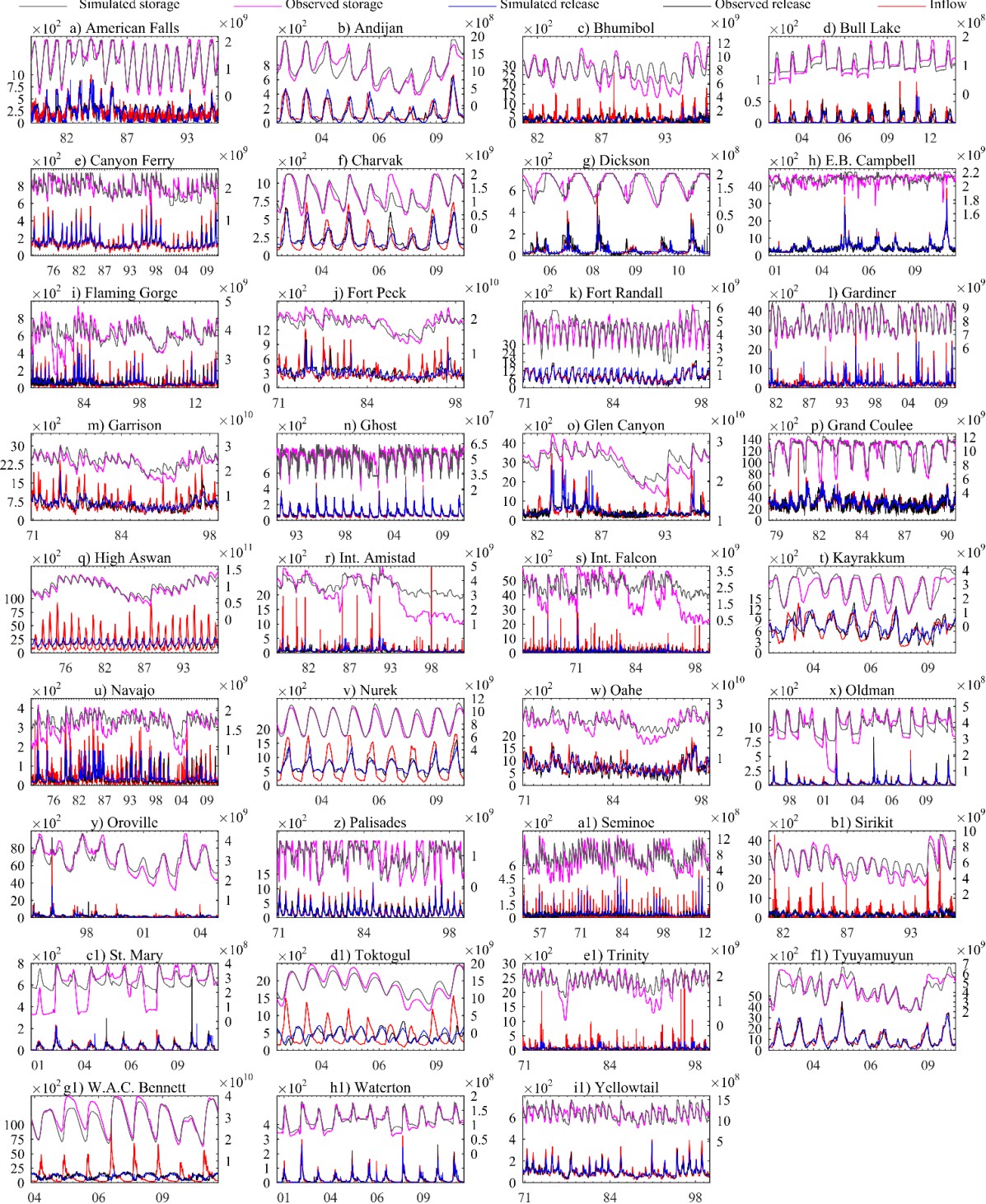

**Figure 6: Daily and monthly reservoir simulations using DZTR model with a generalized parametrization, x-axis shows month/year, the primary y-axis shows release (m³/s) and the secondary y-axis shows storage (m³).**

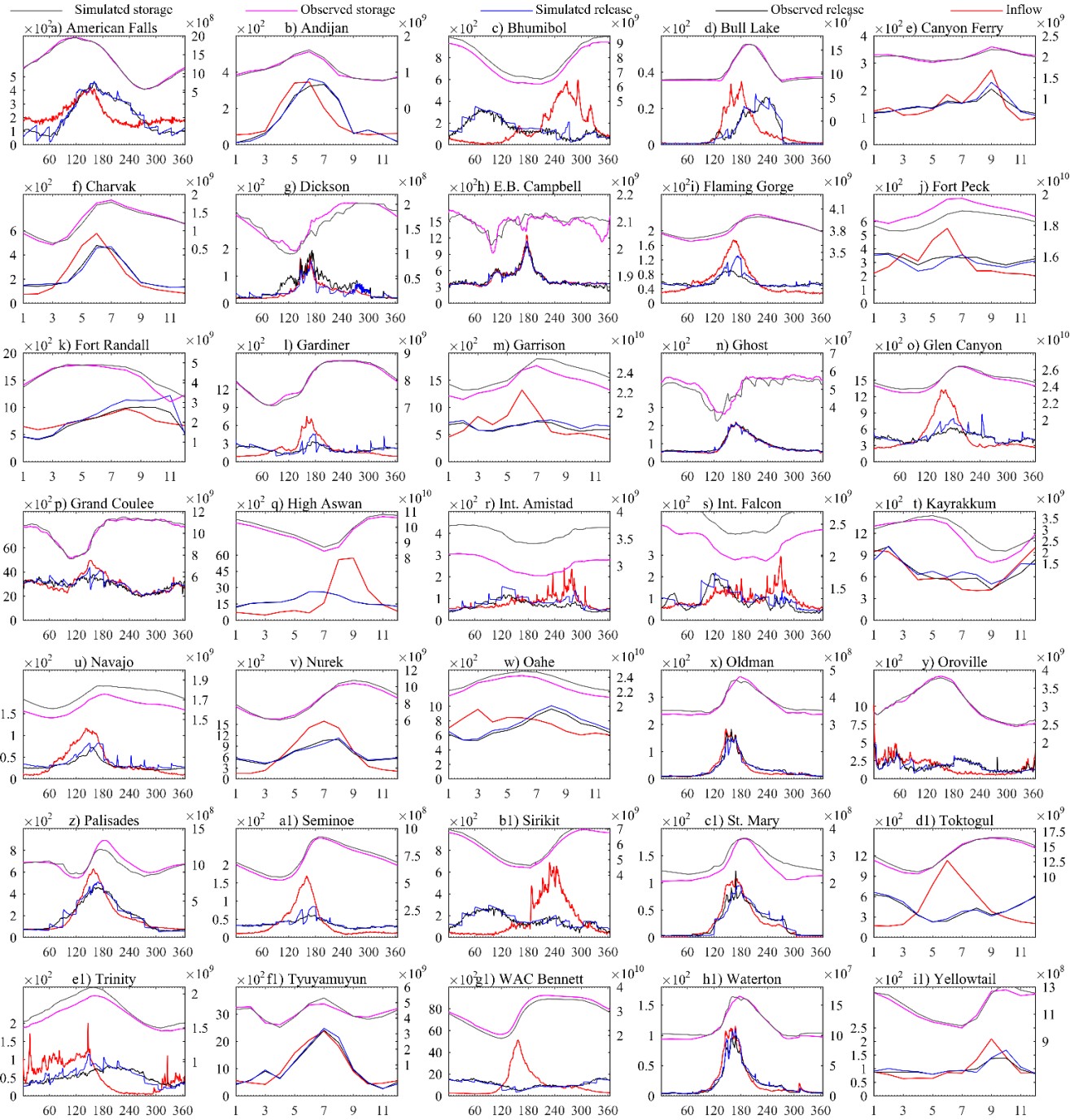

**Figure 7: Long-term average daily or monthly reservoir simulations with generalized parametrization, the x-axis shows days (1-365) or months, (1-12) the primary y-axis shows release (m³/s) and the secondary y-axis shows storage (m³).**

Here we made correction on the comparison result and the figure below is the correct version

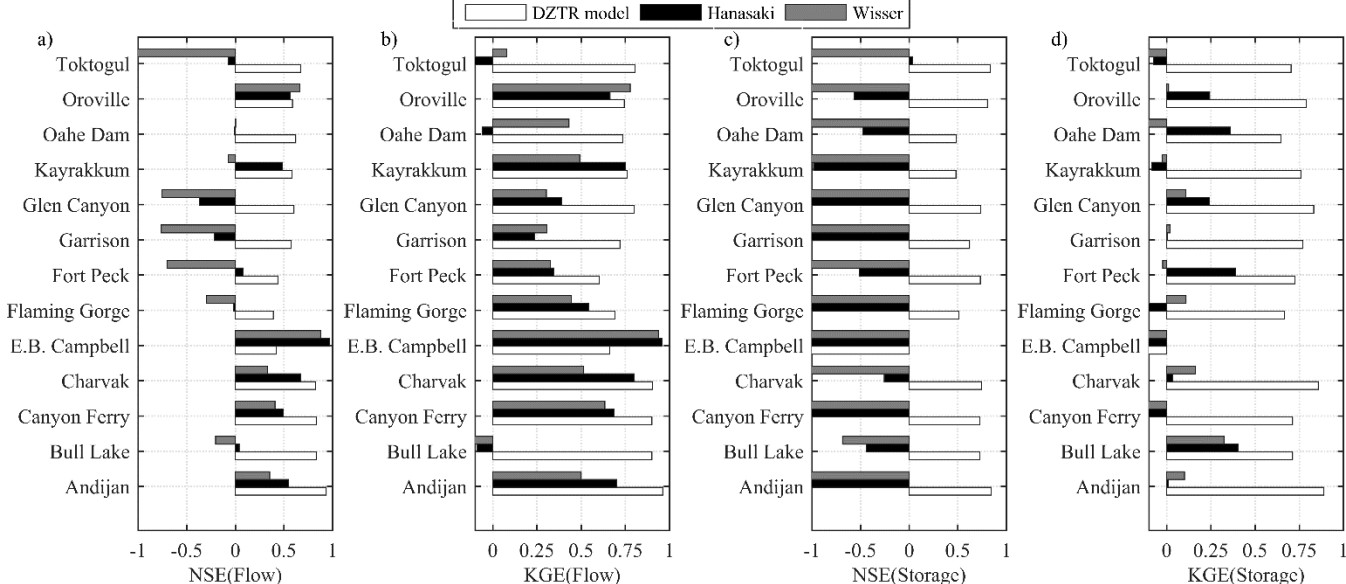

**Figure 8: A comparison of our proposed reservoir operation model with generalized parameters with the models of Hanasaki, et al. (2006) and Wisser et al. (2010): a) NSE(Flow), b) KGE(Flow), c) NSE(Storage), d) KGE(Storage).**

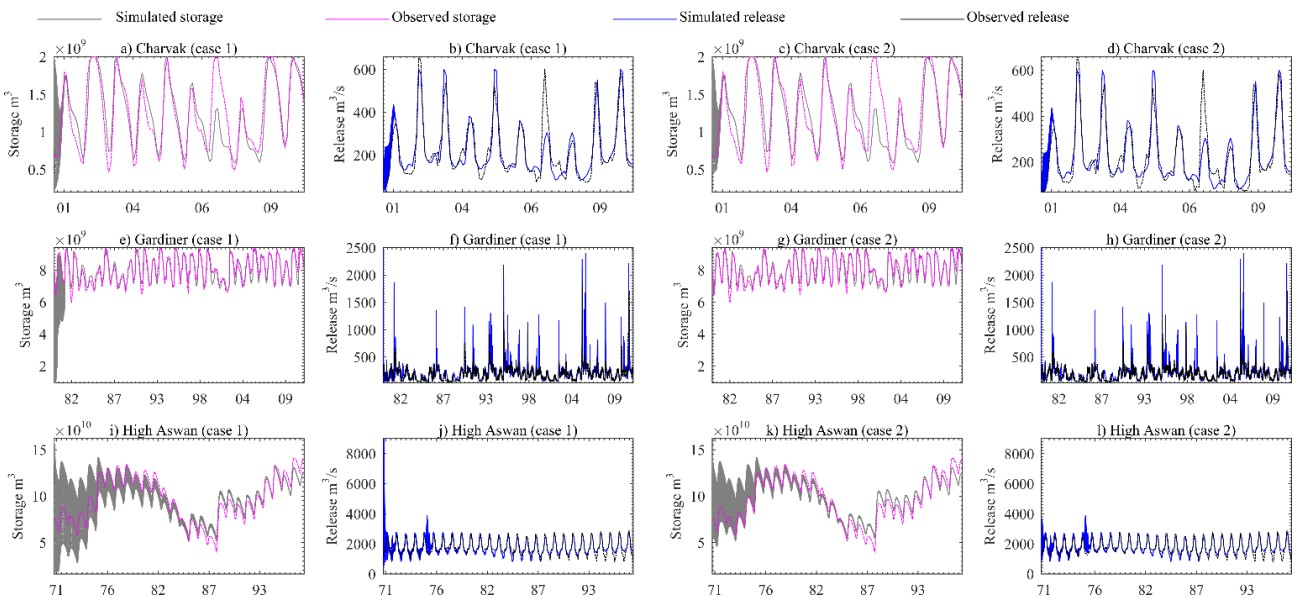

**Figure 9: Reservoir initial storage effect on storage and release simulation: a) Charvak storage case 1, b) Charvak release case 1, c) Charvak storage case 2, d) Charvak release case 2, e) Gardiner storage case 1, f) Gardiner release case 1, g) Gardiner storage case 2, h) Gardiner release case 2, i) High Aswan storage case 1, j) High Aswan release case 1, k) High Aswan storage case 2, l) High Aswan release case 2**

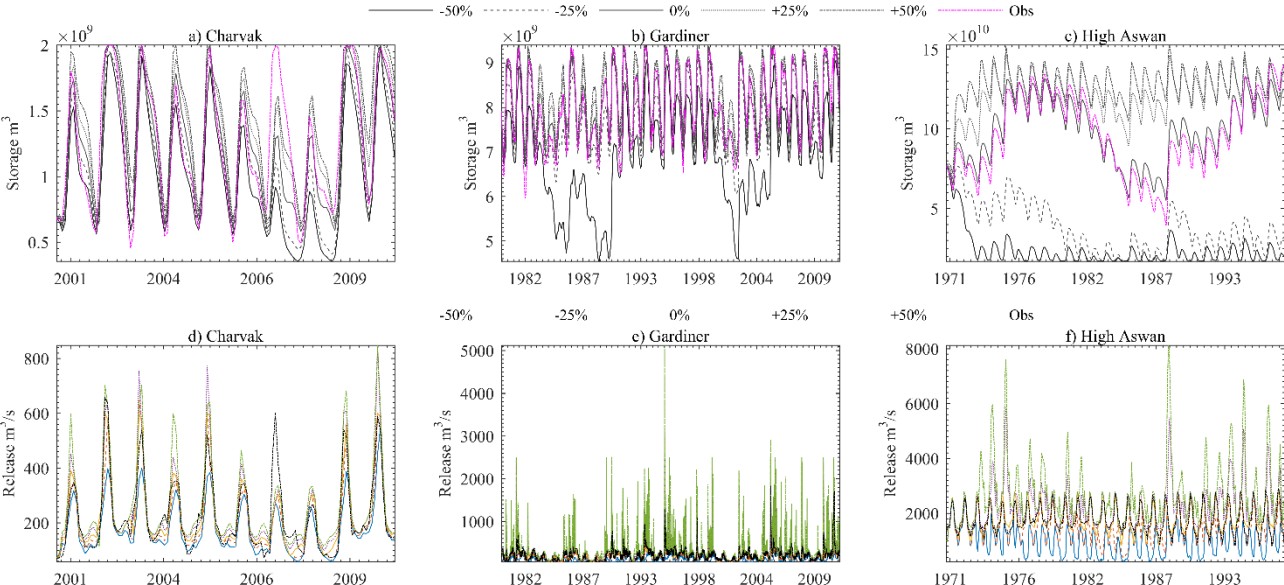

**Figure 10: Inflow bias sensitivity test on storage and release simulation: a) Charvak storage, b) Gardiner storage, c) High Aswan storage, d) Charvak release, e) Gardiner release, f) High Aswan release**

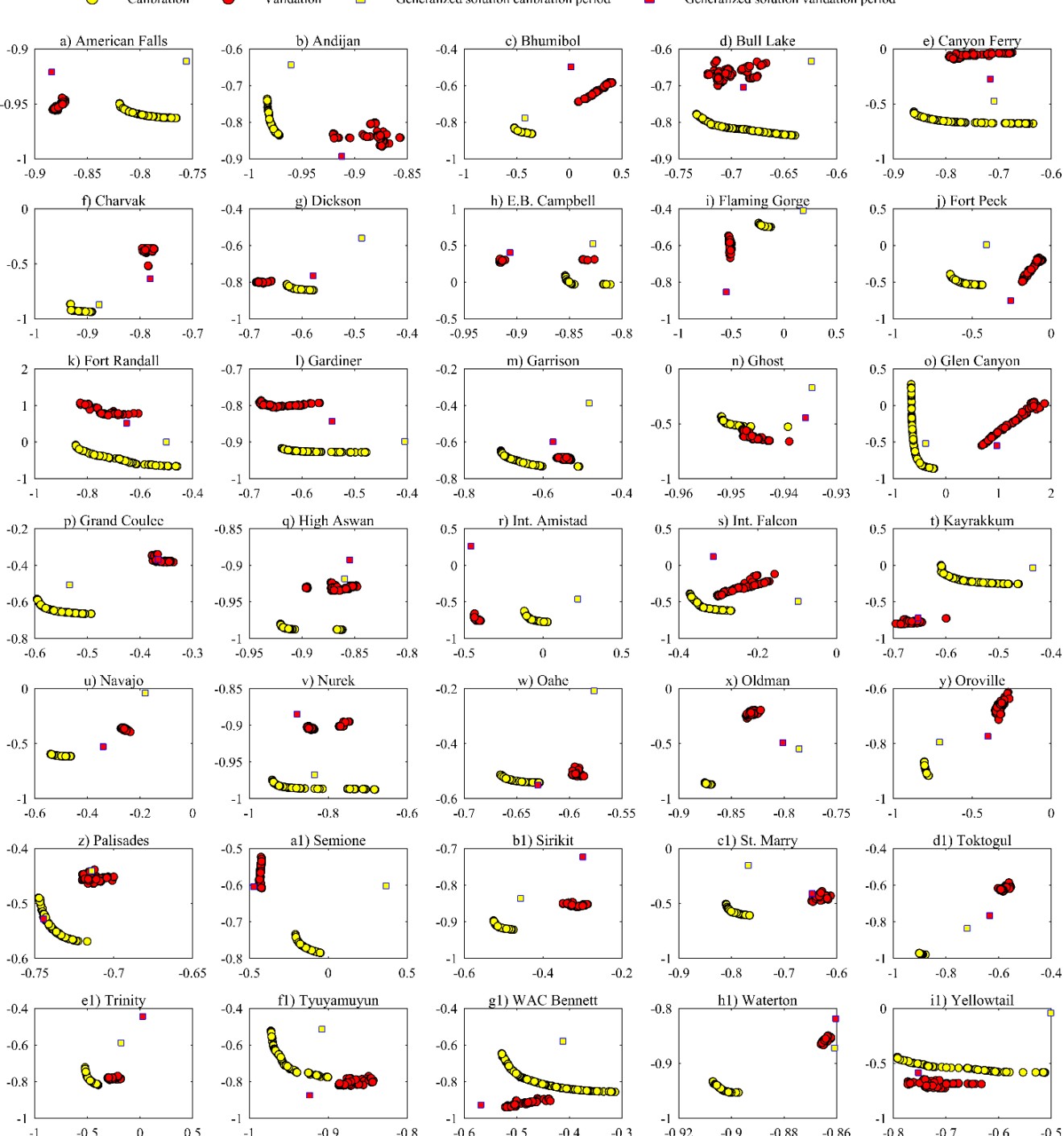

**Figure 11: Reservoir release parameter multi-objective calibration result, x-axis shows NSE (flow) multiplied by -1 and the y-axis shows NSE (storage) multiplied by -1.**

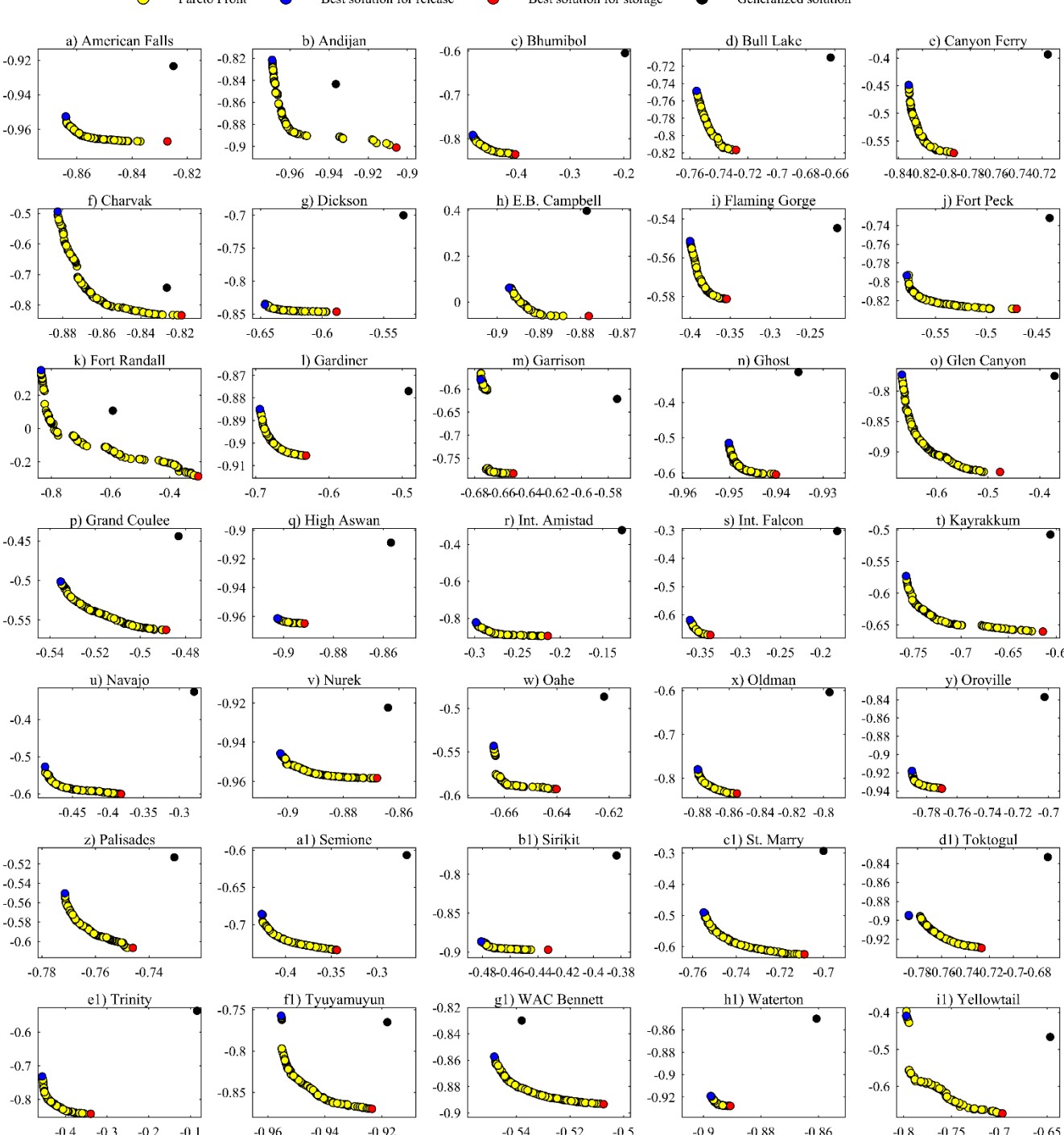

**Figure 12: Reservoir release parameter multi-objective calibration using all available data for each reservoirs, x-axis shows NSE (flow) multiplied by -1 and the y-axis shows NSE (storage) multiplied by -1.**

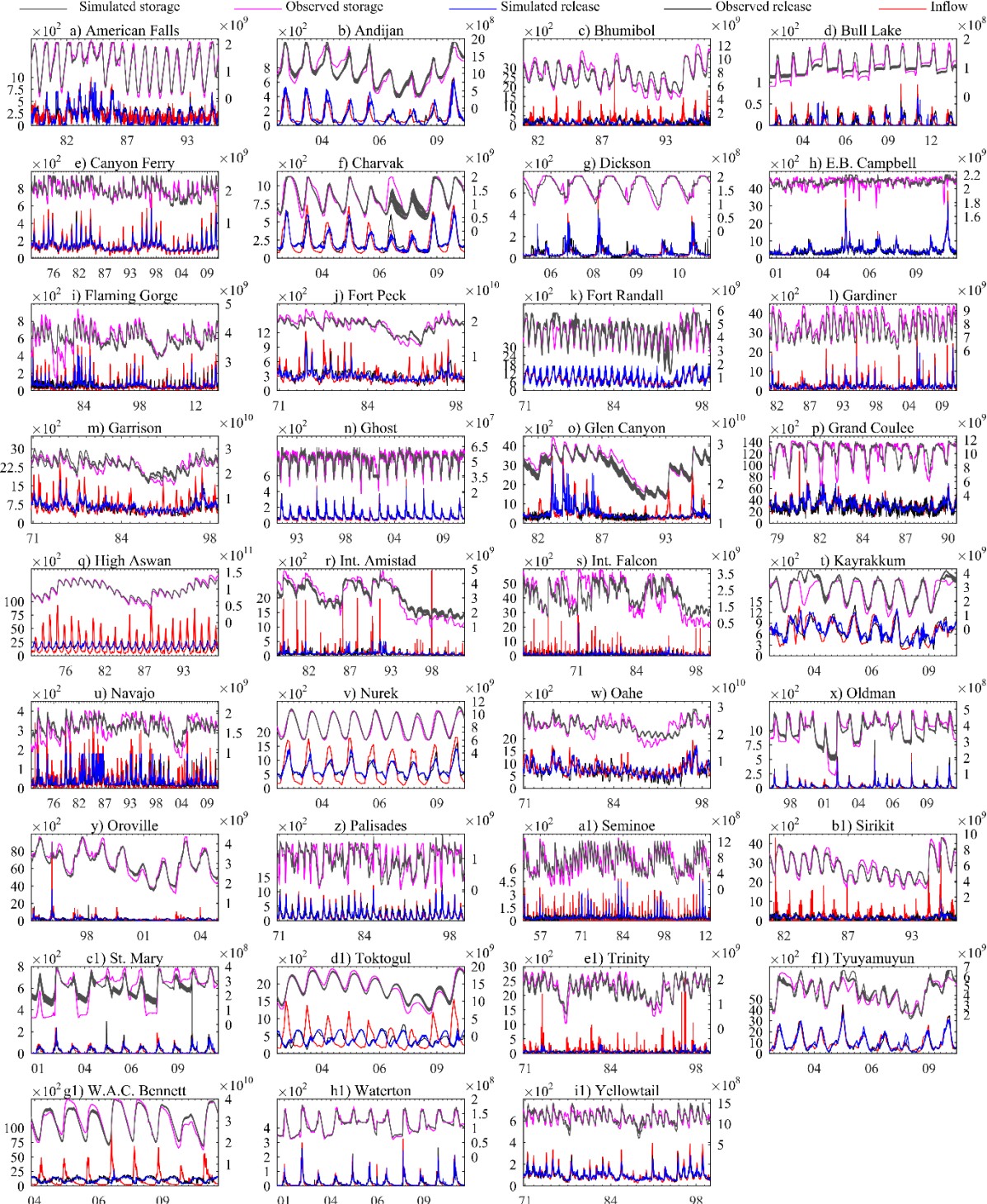

**Figure 13: Daily and monthly reservoir simulations using DZTR model with a generalized parametrization, x-axis shows month/year, the primary y-axis shows release (m³/s) and the secondary y-axis shows storage (m³).**

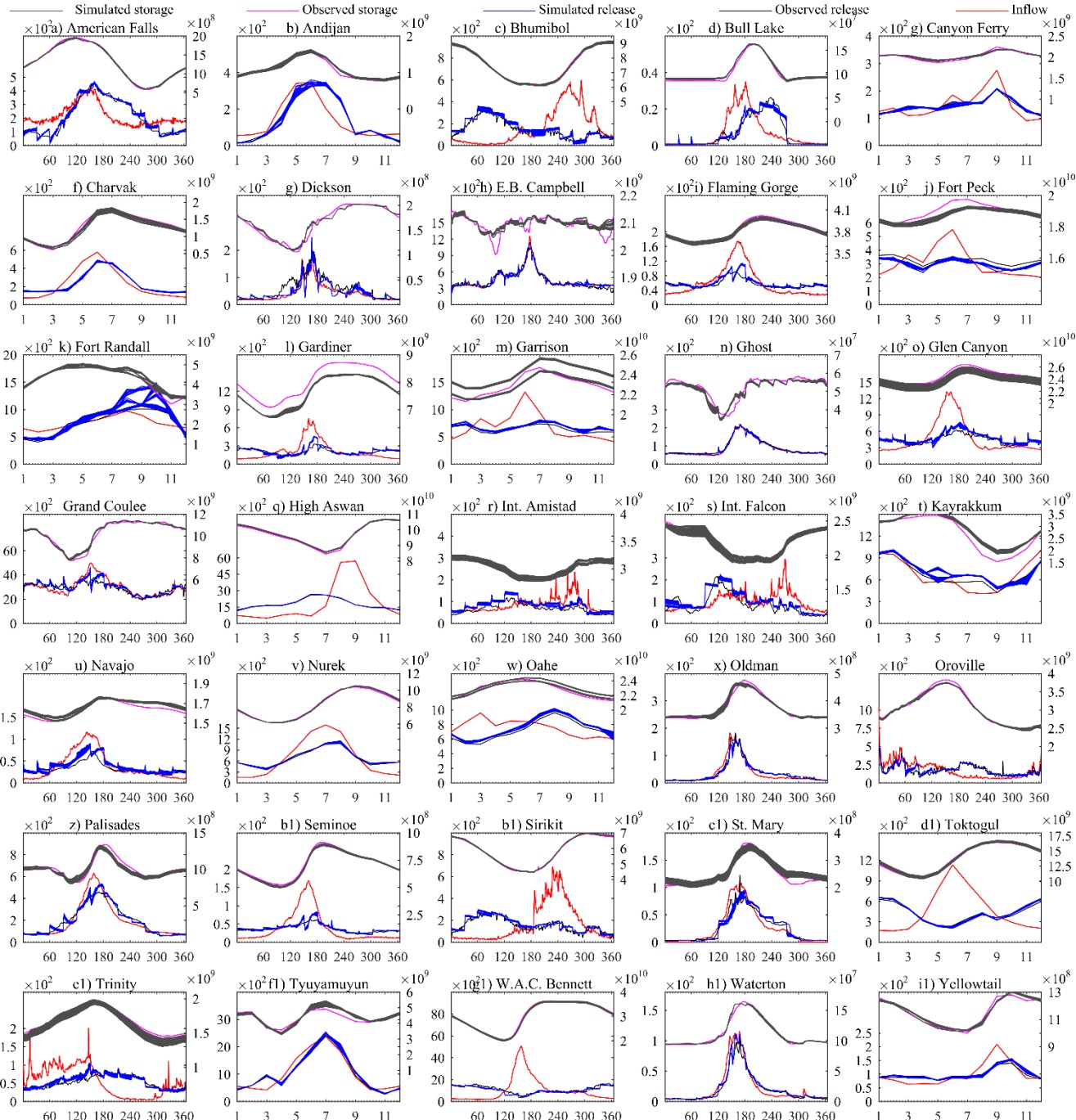

**Figure 14: Long-term average daily or monthly reservoir simulations with generalized parametrization, the x-axis shows days (1-365) or months, (1-12) the primary y-axis shows release (m³/s) and the secondary y-axis shows storage (m³).**

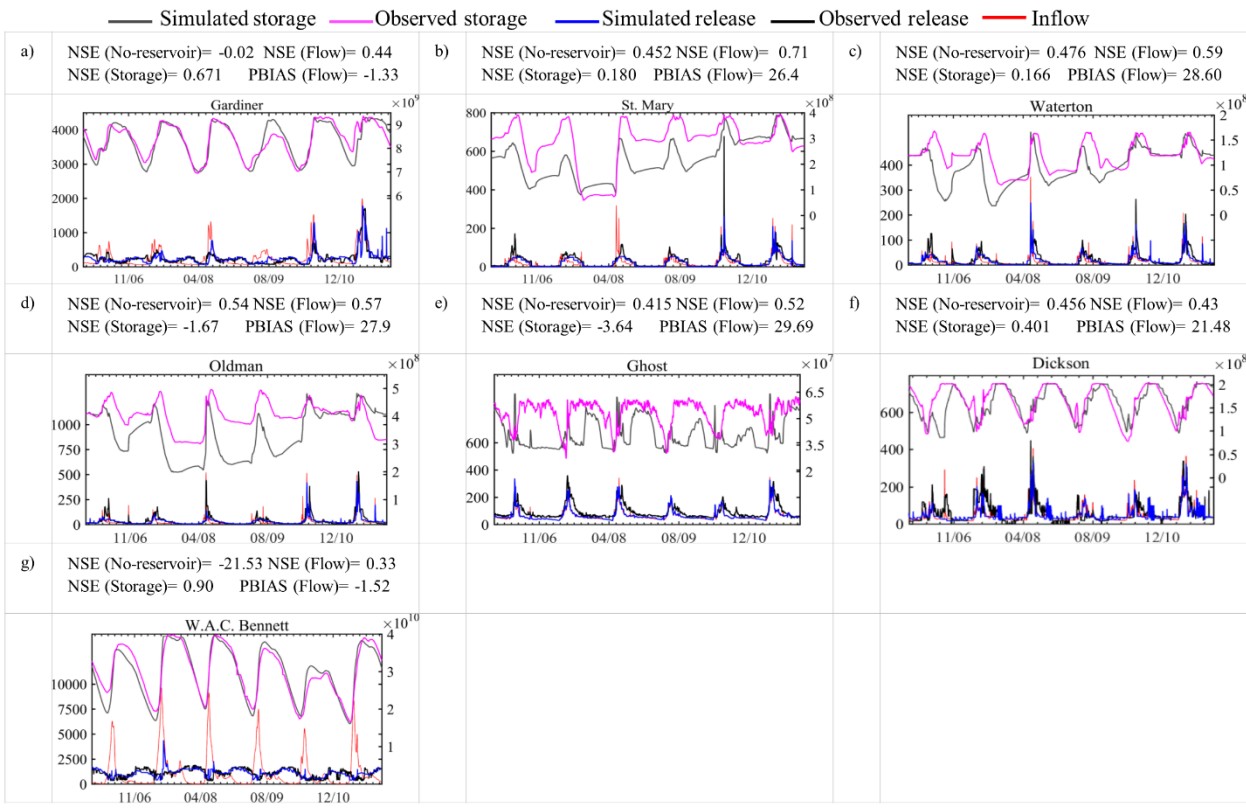

**Figure 15: Reservoir simulation results within MESH model run for selected reservoirs. X-axis shows time (days), the primary y-axis shows release (m³/s) and the secondary y-axis shows storage (m³).**

5   **Table 1: Summary of reservoirs**

| Dam name | Country | Year | Main Purpose | LONG (°) | LAT (°) | Dam height (m) | Capacity (MCM) | $c = \left(\dfrac{Capacity}{MAI}\right)$ | Simulation Period (years) | Percentage Bias (PBIAS) Release vs Inflow |
|---|---|---|---|---|---|---|---|---|---|---|
| American Falls | USA | 1977 | IR | -112.87 | 42.78 | 32 | 2061.5 | 0.303 | 1978-1995 (18) | -3.29 |
| Andijan (y) | Uzbekistan | 1974 | HP | 73.06 | 40.77 | 115 | 1900 | 0.444 | 2001-2013 ᵐ (13) | -0.98 |
| Bhumibol | Thailand | 1964 | IR | 98.97 | 17.24 | 154 | 13462 | 2.645 | 1980-1996 (16) | -10.29 |
| Big Horn | Canada | 1972 | HP | -116.32 | 52.31 | 150 | 1770 | 0.747 | 2002-2011 (10) | 16.08 |
| Bull Lake (y) | USA | 1938 | IR | -109.04 | 43.21 | 24 | 187.2 | 0.883 | 2001-2013(13) | -3.74 |
| Canyon Ferry (y) | USA | 1954 | HP | −111.73 | 46.65 | 69 | 2464.4 | 0.543 | 1971-2011 ᵐ (**40**) | -1.46 |
| Chardara | Kazakhstan | 1968 | IR | 67.96 | 41.24 | 27 | 6700 | 0.354 | 2001-2010 ᵐ (10) | 7.57 |

| Charvak(y) | Uzbekistan | 1977 | HP | 69.97 | 41.62 | 168 | 2000 | 0.284 | 2001-2010 [m] (10) | 1.6 |
|---|---|---|---|---|---|---|---|---|---|---|
| Dickson | Canada | 1983 | WS | -114.21 | 52.05 | 40 | 203 | 0.167 | 2005-2011(6) | 27.3 |
| E.B. Campbell(y) | Canada | 1963 | HP | -103.40 | 53.66 | 34 | 2200 | 0.153 | 2000-2011(12) | -1.69 |
| Flaming Gorge(y) | USA | 1964 | WS | -109.42 | 40.91 | 153 | 4336.3 | 2.460 | 1971-2017(**46**) | -6.37 |
| Fort Peck(y) | USA | 1957 | FC | -106.41 | 48.00 | 78 | 23560 | 2.210 | 1970-1999 [m] (30) | 6.33 |
| Fort Randal | USA | 1953 | FC | -98.55 | 43.06 | 50 | 6683 | 0.240 | 1970-1999[m] (30) | -1.43 |
| Gardiner | Canada | 1968 | IR | -106.86 | 51.27 | 69 | 9870 | 1.460 | 1980-2011(**32**) | -3.44 |
| Garrison(y) | USA | 1953 | FC | -101.43 | 47.50 | 64 | 30220 | 1.436 | 1970-1999 [m] (30) | -5.79 |
| Ghost | Canada | 1929 | HP | -114.70 | 51.21 | 42 | 132 | 0.048 | 1990-2011(22) | 5.43 |
| Glen Canyon(y) | USA | 1966 | HP | -111.48 | 36.94 | 216 | 25070 | 2.230 | 1980-1996(17) | -6.87 |
| Grand Coulee | USA | 1942 | IR | -118.98 | 47.95 | 168 | 6395.6 | 0.124 | 1978-1990(12) | -3.37 |
| High Aswan | Egypt | 1970 | IR | 32.88 | 23.96 | 111 | 162000 | 2.843 | 1971-1997 [m] (26) | -3.34 |
| Int. Amistad | USA/Mexico | 1969 | IR | -101.05 | 29.45 | 87 | 6330 | 2.457 | 1977-2002(25) | -20.28 |
| Int. Falcon Lake | USA/Mexico | 1954 | FC | -99.17 | 26.56 | 53 | 3920 | 1.045 | 1958-2001(43) | -14.48 |
| Kayrakkum(y) | Tajikistan | 1959 | HP | 69.82 | 40.28 | 32 | 4160 | 0.199 | 2001-2010 [m] (10) | 1.19 |
| Navajo | USA | 1963 | IR | -107.60 | 36.80 | 123 | 1278 | 1.744 | 1971-2011(**40**) | -21.07 |
| Nurek | Tajikistan | 1980 | IR | 69.35 | 38.37 | 300 | 10500 | 0.540 | 2001-2010 [m] (10) | 0.28 |
| Oahe Dam(y) | USA | 1966 | FC | -100.40 | 44.45 | 75 | 29110 | 1.244 | 1970-1999 [m] (30) | -5.366 |
| Oldman River | Canada | 1991 | IR | -113.90 | 49.56 | 76 | 490 | 0.446 | 1996-2011(16) | 3.98 |
| Oroville(y) | USA | 1968 | FC | -121.48 | 39.54 | 235 | 4366.5 | 0.804 | 1995-2004(11) | 4.20 |
| Palisades | USA | 1957 | IR | -111.20 | 43.33 | 82 | 1480.2 | 0.242 | 1970-2000(31) | 0.48 |
| Seminoe | USA | 1939 | IR | -106.91 | 42.16 | 90 | 1254.8 | 1.048 | 1951-2013 [m] (**63**) | -4.10 |
| Sirikit | Thailand | 1974 | IR | 100.55 | 17.76 | 114 | 9510 | 1.834 | 1981-1996(16) | -7.32 |
| St. Mary | Canada | 1951 | IR | -113.12 | 49.36 | 62 | 394.7 | 0.492 | 2000-2011(12) | 0.16 |
| Toktogul(y) | Kyrgyzstan | 1978 | HP | 72.65 | 41.68 | 215 | 19500 | 1.393 | 2001-2010 [m] (10) | -6.34 |
| Trinity | USA | 1962 | IR | -122.76 | 40.80 | 164 | 2633.5 | 1.470 | 1970-2000(31) | -4.18 |
| Tyuyamuyun | Turkmenistan | N/A | IR | 61.40 | 41.21 | N/A | 6100 | 0.204 | 2001-2010 [m] (10) | -2.43 |
| W.A.C. Bennett | Canada | 1967 | HP | -122.20 | 56.02 | 183 | 74300 | 1.200 | 2003-2011(9) | 5.41 |
| Waterton | Canada | 1992 | IR | -113.67 | 49.32 | 55 | 172.7 | 0.258 | 2000-2011(12) | -10.34 |
| Yellowtail | USA | 1967 | IR | -107.95 | 45.30 | 160 | 1760.6 | 0.489 | 1970-2000[m] (31) | -1.693 |

[2]Main purpose: **WS**-Water Supply, **HP**-Hydropower **IR**-Irrigation **FC**-Flood Control

(m) Represents monthly data and simulation

(y) Represents multiple reservoir models are compared on this reservoir

**Table 2: Reservoir initial storage effect on storage and release simulation**

| | | Case 1 $S_0= [0.1S_{max} \, S_{max}]$ | | Case 2 $S_0=[min(obs) \, max(obs)]$ obs= observed for all Jan 1[st] | |
|---|---|---|---|---|---|
| | | NSE(Storage) | NSE(Flow) | NSE(Storage) | NSE(Flow) |
| **Charvak** | No spin-up | [0.61 0.74] | [0.79 0.83] | [0.61 0.74] | [0.79 0.83] |
| | 1yr spin-up | [0.74 0.74] | [0.82 0.82] | [0.74 0.74] | [0.82 0.82] |
| **Gardiner** | No spin-up | [-0.43 0.88] | [0.35 0.51] | [0.87 0.88] | [0.44 0.49] |
| | 1yr spin-up | [0.76 0.87] | [0.49 0.51] | [0.87 0.87] | [0.49 0.49] |
| **High Aswan** | No spin-up | [0.38 0.91] | [-0.28 0.85] | [0.42 0.91] | [0.52 0.85] |
| | 1yr spin-up | [0.58 0.91] | [0.62 0.85] | [0.58 0.91] | [0.62 0.85] |

**Table 3: Inflow bias sensitivity test on storage and release simulation**

| | | -50% | -25% | 0% | 25% | 50% |
|---|---|---|---|---|---|---|
| **Charvak** | NSE(Storage) | -1.95 | 0.25 | 0.74 | 0.52 | -0.21 |
| | NSE(Flow) | -0.06 | 0.54 | 0.82 | 0.57 | -0.07 |
| **Gardiner** | NSE(Storage) | -2.00 | 0.74 | 0.88 | 0.79 | 0.66 |
| | NSE(Flow) | -0.21 | 0.47 | 0.49 | -0.43 | -2.02 |
| **High Aswan** | NSE(Storage) | -9.37 | -5.96 | 0.90 | -0.60 | -1.45 |
| | NSE(Flow) | -3.90 | -0.34 | 0.80 | -2.29 | -8.70 |