# Peer review of "Representation and Improved Parameterization of Reservoir Operation in Hydrological and Land Surface Models"

_Hydrology and Earth System Sciences, 2019_

## Referee Comment (RC1) · Anonymous Referee #1 · 19 Feb 2019

The manuscript provides a review of the representation of reservoir operations in a range of hydrology models, then presents a reservoir operations model which explicitly represents storage zones. The optimization scheme AMALGAM is then used to optimize the releases and reservoir zones parameters toward reproducing observed operations. Authors evaluate the models over 37 reservoirs globally and with respect to other previously established reservoir operations schemes. Authors conclude that this explicit representation of storage zones increases the accuracy of representation of reservoir operations. Caveats include the need for data to support the optimization of the operations and the reliance on good calibration of hydrology models to reduce biases in inflow.

The paper is very well written. The introduction summarizes the use of reservoir operations models to complement a range of hydrology models. The explicit representation of reservoir storage zones , optimized/calibrated to match existing reservoir operations is very sound. While the introduction is nicely put together and provides a good review of water management models associated with different scales of hydrology models, it does not support the title. The contribution of the science is mostly in the representation of those new rules. In brief, this model has a very sound and promising concept for reservoir releases, but the models comes out as "oversold" because it lacks the representation of important processes (withdrawals, return flow, dynamic operations etc) and can only be applied on a fraction of reservoirs. A discussion on how it could be implemented in conjunction with existing simplified representations where data is not available, including other driving dynamics such as water withdrawals, would increase the impact of the paper and its leverage by others. As presented, the paper seems to be better suited for a journal presenting geophysical models development and validation, and some clarification of its usage would also be necessary.

1/ Discussion/Contribution to the science

- The specifications for Hanasaki et al. (2006), Haddeland et al. (2006) and other models were specifically to not rely on observed reservoir operations due to the data challenge, and the biases in reservoir inflow estimates. The authors provide some arguments on how satellite data and well calibrated hydrology models are available now. The "why it can be done now" seems to be justified yet the availability and accuracy of those required storage and release observations are not available yet and their accuracy still require research in order to meet water management requirements. This reality decreases the impact of this new model; this data challenge allowed only 37 reservoirs to be represented out of 6000+ reservoir globally with other models.

- The new model is presented to be better for multi objective purposes yet it is compared with reservoirs for flood control mostly (Hanasaki models) because of the lack of water demand information. This is actually a huge deficiency of the new model. All other models explicitly represent not only reservoir operations but also spatially distributed withdrawals and return flow.

Again, the concept is of multiple zones is very sound and appreciated. The valuation of the model as one that can replace existing models which have been looking at drivers of spatio-temporal redistribution of water resources, does not seem adequate nor properly supported.

2/ Technical comments:

- Title is not adequate because the paper is mostly about the new model

- Literature review needs some clarification:

o Note that for catchment model, the inflow is often bias corrected before input into the models.

o RiverWare , MODSIM and OASIS are widely used across the US. Note that all those models require foresight to decide on the reservoir releases. In that context, this is how Haddeland et al. (2006) differs from Hanasaki et al. (2006): Haddeland et al. (2006) also uses foresight to decide on the reservoir releases. How is this new model handling foresight?

o Existing reservoir operations model in a catchment model: Zhao, G., H. Gao, B.S. Naz, S.-C. Kao, N. Voisin, 2016 : Integrating a reservoir regulation scheme into a spatially distributed hydrological model". Advances in Water Resources, 98, 16-31. 2016. doi : 10.1016/j.advwatres.2016.10.014

o Note that Hanasaki et al. (2006) follow one priority use. Voisin et al. (2003) introduced storage-and-release targets toward combining flood control and irrigation, i.e. multi objective use.

o Although non explicit – storage zones were already implicitly represented in Hanasaki et al. (2006) and other models (Wada, Biemans, etc); Spilling when overflowing (i.e. max reservoir at 95% full), and no release when storage gets below 10% of maximum

storage. The contribution is in the explicit representation of those storage zones and their calibration, which, again, is a very sound idea and approach.

- Line 20: the time steps seem off in the equation. Given the time delay between precipitation and when runoff is available to drain, the equation does not seem right for an assumed time step ranging from days to half hours.

- Initial storage and inflow sensitivity section – I am not sure about the information brought up by the sensitivity to initial storage. A significant warm up is always required and is larger for large reservoirs. This model is expected to have storage data available so why are those not used? The use of the section needs some clarification.
* * *

---

## Referee Comment (RC2) · Anonymous Referee #2 · 24 Feb 2019

The paper "Representation of Water Management in Hydrological and Land Surface Models" presents a new scheme for representing reservoir operation in large-scale hydrological and land-surface models. The paper is relevant to HESS readership. It starts by providing a relatively good review of the reservoir operation algorithms both in operational and large-scale models, although several new contributions have missed (please see below just as a sample). The paper is well-written, particularly in the first two sections and the way different algorithms are classified is interesting because it provides a fresh perspective on taxonomy of existing reservoir operation models. The algorithm proposed is simple conceptually and therefore is suitable for the application suggested, although it may end up awfully over-parameterized, in the case of suggested configuration when storage/release thresholds are updated at each month.

[Figure]

This makes the algorithms very limited in scope because the data support for such parametrization is not available in many places of the globe, even in North America despite what mentioned in the paper. Overall, the paper makes a modest contribution to the discussion around representing reservoir operation in large-scale models by providing a new modeling hypothesis, however while the pros of the algorithm is well highlighted, the cons are not really discussed. In addition, I do not believe a new reservoir algorithm, which potentially requires a lot of parameters and cannot represent the dynamics of water withdrawals, can solve the diverse set of grand challenges embedded in "Representation of Water Management in Hydrological and Land Surface Models". As a result, I do agree with the Anonymous Referee #1 that the contribution made is largely oversold. Finally, some of the details in the modeling and results should be better summarized and very important implications, particularly on the trade-off between representing reservoir storage and release, should be better discussed. I suggest the paper undergoes major revisions to address the specific issues raised below:

1) The title should be changed: A new reservoir algorithm cannot solve all problems in representing water management in large-scale models.

2) Although pre-2015 contributions are covered relatively well, new contributions are largely overlooked. Please update the literature review. The contributions named below are just a very limited sample of important new contributions missed in the paper and are given only to help authors to start refurbishing their introduction and framing their algorithm in a wider context:

Pokhrel, Y. N., Hanasaki, N., Wada, Y., & Kim, H. (2016). Recent progresses in incorporating human land–water management into global land surface models toward their integration into Earth system models. Wiley Interdisciplinary Reviews: Water, 3(4), 548-574.

Hanasaki, N., Yoshikawa, S., Pokhrel, Y., & Kanae, S. (2018). A global hydrological simulation to specify the sources of water used by humans. Hydrology and Earth

[Figure]

System Sciences, 22(1), 789.

Ehsani, N., Vörösmarty, C. J., Fekete, B. M., & Stakhiv, E. Z. (2017). Reservoir operations under climate change: storage capacity options to mitigate risk. Journal of Hydrology, 555, 435-446.

Masaki, Y., Hanasaki, N., Biemans, H., Schmied, H. M., Tang, Q., Wada, Y., ... & Hijioka, Y. (2017). Intercomparison of global river discharge simulations focusing on dam operation—multiple models analysis in two case-study river basins, Missouri–Mississippi and Green–Colorado. Environmental Research Letters, 12(5), 055002.

Solander, Kurt C., John T. Reager, Brian F. Thomas, Cédric H. David, and James S. Famiglietti. "Simulating human water regulation: The development of an optimal complexity, climate-adaptive reservoir management model for an LSM." Journal of Hydrometeorology 17, no. 3 (2016): 725-744.

Coerver, H. M., Rutten, M. M., & van de Giesen, N. C. (2018). Deduction of reservoir operating rules for application in global hydrological models. Hydrology & Earth System Sciences, 22(1).

3) Section 3.3: The authors suggest updating the storage/release parameters on the monthly scale to represent the seasonality: So should we end up with 72 parameters for a single reservoir?! Is this something really suitable for using in the context of large-scale models that have already a lot of parameters and face with scarce and low quality observations particularly in terms of human-water interactions? Because of being heavily over-parameterized, this scheme is only suitable where there are at least multiple years of continuous and high quality data available: Even in North America, such data availability is widely limited considering the discontinuity in in-situ measurements of storage and release across regional reservoir networks even in western Canada and US, where most of the case studies of this work are located. The fact that many large dams are privately owned and therefore the data are not publicly available is not mentioned anywhere in the paper: This is the particularly the case of large hydroelectric dams in US, Canada and Brazil that together account for large proportion of annual reservoir storage globally. Please discuss properly this important issue of the scheme along with other limitations of the proposed model at least with the same weight as its strengths. Highlighting the limitation of the proposed algorithm must be a key consideration during revisions.

4) Section 4.1: What are the uncertainties in the generalized parameterizations? The percentiles corresponding to monthly target storage and release should be different for different reservoirs and I can imagine that it might be several combinations of percentiles that can provide similar modeling efficiency even in one single reservoir: Please discuss and provide some evidence on the uncertainty in these generalized parameterizations.

5) Figure 11 shows an explicit trade-off between reservoir release and reservoir storage during calibration: This means that it is impossible to reach the skill in representing each objective function without compromising on the other, implying that the algorithm is unable to track both reservoir release and storage optimally at the same time: Isn't it a limitation in the model? How much this uncertainty contributes into uncertainty in identifying the role of reservoir in modifying the natural streamflow regime? This very important point seems to be wholly ignored at this stage and should be addressed in revisions.

6) Figure 11 again: It is surprising that the results during validation do not show the trade-off observed during calibration in several reservoirs: Doesn't this show that the parametrization is very sensitive to the period used for parameter identification? Also, the results during calibration are non-dominated by definition; however, do the results during validation also remain non-dominated when compared with other possible parametrizations that have been dominated during the calibration? The sensitivity of model parameters to training data and the robustness of results during validation should be well discussed during the revision and supported by experimental results.

7) Incorporation of the algorithms in the considered large-scale model seems to be limited to one reservoir at the time. Whereas in real cases, multiple reservoirs are built over one river and therefore the cal/val procedure and the skill of the reservoir algorithm should be tested when the outflow from one reservoir is the inflow to the next reservoir. The paper ignores this as many other similar contributions do. But I believe this is worth at least proper discussion because the challenge is out there and has remained, indeed, unsolved. Up to the time that the problem of considering multiple reservoirs in one basin is not properly solved, the results of large-scale models remain only as naive simulations of a virtual hydrologic reality at the basin-scale, which contributes to a huge uncertainty at regional, continental and global scales.

––––––––––––––––––––––

---

## Author Comment (AC1) · 13 May 2019

The manuscript provides a review of the representation of reservoir operation in a range of hydrology models, then presents a reservoir operation model which explicitly represents storage zones. The optimization scheme AMALGAM is then used to optimize the releases and reservoir zones parameters toward reproducing observed operation. Authors evaluate the models over 37 reservoirs globally and with respect to other previously established reservoir operations schemes. Authors conclude that this explicit representation of storage zones increases the accuracy of representation of reservoir operation. Caveats include the need for data to support the optimization of the operations and the reliance on good calibration of hydrology models to reduce biases in inflow.

We would like to thank the reviewer for the time spent to carefully review our manuscript. We greatly appreciate the important points raised. We present our response to reviewer's comments below. The reviewer comments are listed below in regular, black text, and our response in regular blue text.

The paper is very well written. The introduction summarizes the use of reservoir operation models to complement a range of hydrology models. The representation of reservoir storage zones, optimized/calibrated to match existing reservoir operation is very sound. While the introduction is nicely put together and provides a good review of water management models associated with different scales of hydrology models, it does not support the title. The contribution of the science is mostly in the representation of those new rules. In brief, this model has a very sound and promising concept for reservoir releases, but the models comes out as "oversold" because it lacks the representation of important processes (withdrawals, return flow, dynamic operations etc.) and can only applied on a fraction of reservoirs. A discussion how it could be implemented in conjunction with existing simplified representation where data is not available, including other driving dynamics such as water withdrawals would increase the impact of the paper and its leverage by others. As presented, the paper seems to be better suited for a journal presenting geophysical models development and validation, and some clarification of its usage would be necessary.

Accurate representation of water management is still a major challenge for land surface/hydrology models. In this study, we focused on reservoir representation as one of the core water management

components regulating streamflow and our aim and contribution was to introduce an improved parameterization for reservoir operation under the condition of sufficient data availability. The minimum requirements for data are time series of reasonable length for reservoir storage and release. Finally we integrate the reservoir model into a land surface model.

We agree that the title is much wider than the objective and contribution of our work and, therefore, we suggest changing the title to: "Representation and Improved Parameterization of Reservoir Operation in Hydrological and Land Surface Models".

Since the reviewer comments will not be addressed by changing the title only, we would like to improve the manuscript by adding discussion on the following issues as suggested by the reviewer: 1) How to integrate and use the presented reservoir operation scheme with other existing reservoir operation schemes that requires less data. 2) How to link different component of water management such as water withdrawals for irrigation, return flow, prioritization among competing demands.

The responses and clarifications given below include the scope and direction of the discussion and clarification we have been adding to the revised manuscript.

1 Discussion/contribution on the science

The specification for Hanasaki et al. (2006), Haddeland et al. (2006) and other models were specifically to not rely on observed reservoir operation due to the data challenge, and the biases in reservoir inflow estimates. The author provide some arguments on how satellite data and well calibrated hydrology models are available now. The "why it can be done now" seems to be justified yet the availability and accuracy of those required storage and release observation are not available yet and their accuracy still require research in order to meet water management requirements. This reality decrease the impact of this new model; this data challenge allowed only 37 reservoirs to be represented out of 6000+ reservoir globally with other models.

Thanks for raising this point. As demonstrated in section 2, the methods of Hanasaki et al., (2006) and Haddeland et al. (2006) and other reservoir models derived from these two require no major data. Due to limitation of data, we are not suggesting by any means to fully replace existing methods such as Hanasaki et al. (2006) with the presented reservoir model, but instead they can be used together to make better use of the data when they exist.

The challenge we are trying to highlight is that there is a lack of reservoir sub-models within land surface and large scale hydrological models that can potentially utilize existing data to appropriately represent reservoir operation across storage zones. We agree that the data requirement limits the application of DZTR at global scale, but as demonstrated in our results, for reservoirs with only limited data support, the results of DZTR method are superior to the existing

ones in terms of simulating both reservoir storage and releases. Further, in the case where inflow data are not available, the generalized parameterization can be used effectively and gave generally better results than other reservoir models at the scale of application. Part of the discussion in the revised manuscript is directed to show how to effectively use the presented (DZTR method and parameterization) approach in case of data unavailability as detailed in the coming paragraphs.

One approach is to integrate our proposed method in land surface/catchment models along with other reservoir operation methods (e.g., Hanasaki et al., 2006). Then, within the land surface/catchment models, identifier flags can be used to indicate which method applies to which reservoirs. The DZTR approach can only be activated for reservoirs with data support, while the remaining reservoirs can use other approaches as dictated by data availability. We have been following such an implementation within the MESH model. As shown in our results, reservoir regulation has a huge impact on downstream flows if the reservoir is highly regulated and/or is of multi-year type (c>0.5). Thus, more emphasis can be put on those reservoirs with c>0.5.

At the moment, such methods will be more effective at the regional than global scale (for example for Saskatchewan River Basin in our case), because modellers at regional scale have better access to storage-inflow-outflow data and have better understanding of the system to acquire the necessary reservoir data. In a land surface hydrologic model, important reservoirs are those causing large changes on the downstream flows and those tend to be the larger ones with generally better data availability.

We collected data for 37 reservoirs, as examples, to assess the scheme and showed that the generalized parameterization performed better than the other methods. Data on reservoir storage, inflow and release exist for most reservoirs but sometimes they are not made publically available. The generalized parameterization requires storage and release data. Storage data can be obtained from water level data which is generally available for major reservoirs, and release data can be deduced from the nearest downstream station. Data on reservoir water levels can be easily converted to storage as mentioned at the end of Section 2 in the manuscript. Initiatives are needed to gather and archive such reservoir datasets and move beyond information on reservoir characteristics that is currently available in databases (e.g. GRanD database - Lehner et al., 2011). One of our recommendation is that the target release and storage data be archived for public use at least for highly regulated and multi-year type dams (c>0.5). We intend to expand the discussions on this in the revised manuscript.

The possibility of estimating storage and release data from different satellite data products was mentioned in the manuscript to highlight an optimistic view that such types of data will be more available in the future for successful expansion of use of methods like the presented reservoir operation (optimized or generalized). More recently, Busker et al. (2019) showed an estimation of

volume for 130 reservoirs using surface water dataset and satellite altimetry, which is encouraging. These will be discussed further in the revised manuscript.

The new model is presented to be better for multi objective purposes yet is compared with reservoirs for flood control mostly (Hanasaki models) because of the lack of water demand information. This is actually huge deficiency of the new model. All other models explicitly represent not only reservoir operation but also spatially distributed withdrawals and return flow.

Again, the concept is of multiple zones is very sound and appreciated. The valuation of the model as one that can replace existing models which have been looking at drivers of spatio-temporal redistribution of water resources, does not seem adequate nor properly supported.

Water demand data is needed to use the Hanasaki et al. (2006) method for irrigation dams. In the case of the DZTR approach, the idea is that the DZTR model operates in such a way to infer existing operational rules which cater for those demands. Thus, the release from DZTR accounts, implicitly, for downstream demands as per the intended purpose of the reservoir whether it is for flood control, irrigation, hydropower, etc. or any combination of these. The case study dams in our study include reservoirs with different purposes as shown in Table 1 (reservoirs summary). The DZTR approach showed good performance for reservoirs with different purposes.

If the reservoir purpose is irrigation, the target releases from DZTR are to satisfy irrigation demands because the parameterization is optimized based on observed releases. The release from an irrigation dam will be available for abstraction at the predefined abstraction points downstream of the dam. The abstraction and distribution are separate modules within our land surface model (not discussed in the original manuscript) which take care of (1) actual irrigation demand for the dependant an irrigation area, (2) water abstraction from defined abstraction point along the river below the dam and (3) distribution across the irrigation fields. Regarding the return flow, the excess water flows from the irrigation areas are assumed to join the nearest stream (the grid cell each irrigation tile belongs to). These modules are currently under investigation within the MESH framework which we used as an example to show how the DZTR model can be integrated with hydrological land surface models. However, the paper only focuses on reservoirs and the title will changed to reflect that. Thus, these issues are out of the paper scope and can be only touched upon briefly in the revised manuscript.

In case of multi-purpose reservoirs, e.g. a reservoir that is used simultaneously for hydropower generation, irrigation water supply, and flood control (e.g. High Aswan Dam in Egypt which is one of the studied reservoirs), the DZTR provides the release based on the inflow, and storage conditions and that will be available for irrigation downstream. Hydropower does not consume water but returns it back to the river (except in rare cases where it returns to a different channel).

Flood control is already accounted for in the scheme and becomes relevant when storage is within the flood storage zone. The flexible formulation of DZTR allows to implicitly change the priority for selected time periods (e.g. months or seasons) by changing the target storage values during flood periods (e.g. the storage target before the onset of snowmelt). During these flood months, lowering the target storage would increase the buffer for flood control. Conversely increasing the target storage during other months would be desirable to store water and release during irrigation months. When the scheme is optimized using inflow, release, and storage data, the parameterizations capture these priorities implicitly as expressed in the data. When inflow data are lacking, the generalized parametrization will set the storage zones based on the suggested exceedance probabilities (that were based on all reservoirs used in the study) and the priorities can be assumed as pre-defined. These points are being elaborated in the discussions in the revised manuscript.

2 Technical comments:

Title is not adequate because the paper is mostly about the new model

We agree that the title is misleading. We, therefore, suggest to use the following title for our revised manuscript: "Representation and Improved Parameterization of Reservoir Operation in Hydrological and Land Surface Models".

Literature review needs some clarification

We will expand the review to include more recent work as directed by reviewer #2.

Note that for catchment model, the inflow is often bias corrected before input into the models.

Our understanding is that a catchment model simulates the inflow. In some cases bias correction is applied to precipitation or other climatic variables. In operational models, inflows may be corrected by data assimilation, but these types of models are generally more detailed and case specific. These are not the target for our DZTR reservoir model.

In cases where bias correction of inflow to reservoir is needed, it is possible to achieve it through introducing inflow multiplier parameter with in the reservoir algorithm to adjust the inflow with constant multiplier factor.

RiverWare, MODSIM and OASIS are widely used across the US. Note that all those models require foresight to decide on the reservoir releases. In that context, this is how Haddeland et al. (2006) differs from Hanasaki et al. (2006): Haddeland et al. (2006) also uses foresight to decide on the reservoir releases. How is this new model handling foresight?

We appreciate this information. The models mentioned are detailed water management models that can include explicit operating rules which are not usually available. Our scheme attempts to

infer those rules from the inflow (if available), storage, and release data which are more readily available. The manuscript is being updated to clarify the difference between Haddeland et al. (2006) and Hanasaki et al. (2006).

Regarding handling foresight, the manuscript is being revised to clarify how the DZTR approach handles the foresight as:

The target release does not use long-term forecasts to decide the operational year inflows (e.g. dry year and wet year). Instead, it uses the simulated reservoir storage value to determine the zone to use to calculate the release. That means the operation with multiple zones helps buffer the dry and wet year inflows. For a dry year, the release is automatically reduced as storage will be lower due to less inflows. Conversely, it will increase for wet year as the higher inflows will increase the storage and even move it to a different zone. However, in reality some more preparation is taken to determine dry and wet years. For instance in cold regions, the snow water equivalent upstream of the dam is often monitored to determine the inflow forecast, prioritize release and adjust the water sharing among different sectors. The DZTR scheme is aimed to be included in land surface models to mimic reservoir operation in scenario simulations, not in operational models to aid the decisions on reservoir operation.

Existing reservoir operations model in a catchment model: Zhao, G., H. Gao, B.S. Naz, S.-C. Kao, N. Voisin, 2016: Integrating a reservoir regulation scheme into a spatially distributed hydrological model". Advances in Water Resources, 98, 16-31. 2016. doi : 10.1016/j.advwatres.2016.10.014

Note that Hanasaki et al. (2006) follow one priority use. Voisin et al. (2003) introduced storage-and-release targets toward combining flood control and irrigation, i.e. multi objective use.

Although non explicit – storage zones were already implicitly represented in Hanasaki et al. (2006) and other models (Wada, Biemans, etc); Spilling when overflowing (i.e. max reservoir at 95% full), and no release when storage gets below 10% of maximum storage. The contribution is in the explicit representation of those storage zones and their calibration, which, again, is a very sound idea and approach.

Thanks for these points with which we fully agree. The idea of having zones originates from the classic methods for reservoir design and operation. Our approach attempted to make it as flexible as possible by increasing the number of zones and defining them dynamically. While the dead storage zone is usually physically constrained by the dam design, the flood control zone is subject to management decisions based on several considerations. Release from other intermediate zones are related to demands. We compared our approach to Hanasaki et al. (2006) because their releases are strongly determined based on demands for different sectors, inflow, and forecasted inflow. The only reason we decided to put the approach Haddeland et al. (2006), Hanasaki et al. (2006) and

their derivatives is that, the releases are strongly determined based on demands for different sectors, inflow, and forecasted inflow. Clarifications along those lines are being included in the revised manuscript.

Line 20: the time steps seem off in the equation. Given the time delay between precipitation and when runoff is available to drain, the equation does not seem right for an assumed time step ranging from days to half hours.

The precipitation and evaporation terms in Equation (1) are the direct quantities over the reservoir, not those over the catchment draining into the reservoir. Runoff reaching the reservoir is simulated by a hydrological model of the upstream catchment that would account for delays in the precipitation-runoff generation and routing. We regret that this point was not clear in the original manuscript, and the manuscript is being revised to reflect those clarifications.

Initial storage and inflow sensitivity section – I am not sure about the information brought up by the sensitivity to initial storage. A significant warm up is always required and is larger for large reservoirs. This model is expected to have storage data available so why are those not used? The use of the section needs some clarification.

The initial storage at the beginning of the simulation is an input that needs to be specified to the model. As you mention, the initial values can be prescribed form the observations if available. However, the simulation of a hydrological/land surface model could start at any different time period where there is no observation to prescribe, such as in the case of a future scenario simulation or a hypothetical historical scenario. Additionally, in a long-term simulation, the initial storage may result from a previous model simulation and may not be as close to observations as desired. The aim of the experiment is to examine and show to what extent the initial storage value affects the simulation performance. The outcome at least shows it is better to start with any historical observation for the same starting month of the simulation as well as to what extent the warm-up period needed for large vs smaller reservoirs. The text in the manuscript is being modified accordingly to clarify the purpose of sensitivity to initial storage. We acknowledge that it is well known that longer warm-ups are needed for larger reservoirs and the analysis can be used as a guide for the required warm-up period when regressed against the storage capacity. The application of the scheme in a large scale modelling framework can benefit from those guidelines.

References

Busker, T., de Roo, A., Gelati, E., Schwatke, C., Adamovic, M., Bisselink, B., Pekel, J.-F. and Cottam, A.: A global lake and reservoir volume analysis using a surface water dataset and satellite altimetry, Hydrol. Earth Syst. Sci., 23(2), 669–690, doi:10.5194/hess-23-669-2019, 2019.

Liebe, J., van de Giesen, N. and Andreini, M.: Estimation of small reservoir storage capacities in a semi-arid environment: A case study in the Upper East Region of Ghana, Phys. Chem. Earth, Parts A/B/C, 30(6–7), 448–454, doi:10.1016/J.PCE.2005.06.011, 2005.

Lehner B, Liermann CR, Revenga C, Vörösmarty C, Fekete B, Crouzet P, Döll P, Endejan M, Frenken K, Magome J, et al. 2011. High-resolution mapping of the world's reservoirs and dams for sustainable river-flow management. *Frontiers in Ecology and the Environment* **9** (9): 494–502 DOI: 10.1890/100125

---

## Author Comment (AC2) · 13 May 2019

The paper "Representation of Water Management in Hydrological and Land Surface Models" presents a new scheme for representing reservoir operation in large-scale hydrological and land-surface models. The paper is relevant to HESS readership. It starts by providing a relatively good review of the reservoir operation algorithms both in operational and large-scale models, although several new contributions have missed (please see below just as a sample). The paper is well-written, particularly in the first two sections and the way different algorithms are classified is interesting because it provides a fresh perspective on taxonomy of existing reservoir operation models. The algorithm proposed is simple conceptually and therefore is suitable for the application suggested, although it may end up awfully over-parameterized, in the case of suggested configuration when storage/release thresholds are updated at each month.

This makes the algorithms very limited in scope because the data support for such parametrization is not available in many places of the globe, even in North America despite what mentioned in the paper. Overall, the paper makes a modest contribution to the discussion around representing reservoir operation in large-scale models by providing a new modeling hypothesis, however while the pros of the algorithm is well highlighted, the cons are not really discussed. In addition, I do not believe a new reservoir algorithm, which potentially requires a lot of parameters and cannot represent the dynamics of water withdrawals, can solve the diverse set of grand challenges embedded in "Representation of Water Management in Hydrological and Land Surface Models".

As a result, I do agree with the Anonymous Referee #1 that the contribution made is largely oversold. Finally, some of the details in the modeling and results should be better summarized and very important implications, particularly on the trade-off between representing reservoir storage and release, should be better discussed. I suggest the paper undergoes major revisions to address the specific issues raised below:

We would like to thank the reviewer for all the helpful comments and for the time spent to carefully review our manuscript. We present our response to the reviewer's comments below. The reviewer comments are listed below in regular, black text, and our response in regular blue text.

We appreciate the above important point raised by the reviewer. Since each of the reviewer's above points are separately expanded below, our response addresses the above point in the appropriate section of the numbered list below.

1) The title should be changed: A new reservoir algorithm cannot solve all problems in representing water management in large-scale models.

We agree with the reviewer that the original title was misleading, and as per the objective and contribution of our work, we are suggesting the following title for our revised manuscript: "Representation and Improved Parameterization of Reservoir Operation in Hydrological and Land Surface Models".

2) Although pre-2015 contributions are covered relatively well, new contributions are largely overlooked. Please update the literature review. The contributions named below are just a very limited sample of important new contributions missed in the paper and are given only to help authors to start refurbishing their introduction and framing their algorithm in a wider context:

Pokhrel, Y. N., Hanasaki, N., Wada, Y., & Kim, H. (2016). Recent progresses in incorporating human land–water management into global land surface models toward their integration into Earth system models. Wiley Interdisciplinary Reviews: Water, 3(4), 548-574.

Hanasaki, N., Yoshikawa, S., Pokhrel, Y., & Kanae, S. (2018). A global hydrological simulation to specify the sources of water used by humans. Hydrology and EarthSystem Sciences, 22(1), 789.

Ehsani, N., Vörösmarty, C. J., Fekete, B. M., & Stakhiv, E. Z. (2017). Reservoir operations under climate change: storage capacity options to mitigate risk. Journal of Hydrology, 555, 435-446.

Masaki, Y., Hanasaki, N., Biemans, H., Schmied, H. M., Tang, Q., Wada, Y., ... & Hijioka, Y. (2017). Intercomparison of global river discharge simulations focusing on dam operation at multiple models analysis in two case-study river basins, Missouri–Mississippi and Green–Colorado. Environmental Research Letters, 12(5), 055002.

Solander, Kurt C., John T. Reager, Brian F. Thomas, Cédric H. David, and James S. Famiglietti. "Simulating human water regulation: The development of an optimal complexity, climate-adaptive reservoir management model for an LSM." Journal of Hydrometeorology 17, no. 3 (2016): 725-744.

Coerver, H. M., Rutten, M. M., & van de Giesen, N. C. (2018). Deduction of reservoir operating rules for application in global hydrological models. Hydrology & Earth System Sciences, 22(1).

We thank the reviewer for the suggestions. Some of these papers were not included in the original manuscript, because their contribution about reservoir operation was minimal; however, we are

currently examining the suggested references and some other more recent papers to be included in the revised manuscript.

3) Section 3.3: The authors suggest updating the storage/release parameters on the monthly scale to represent the seasonality: So should we end up with 72 parameters for a single reservoir?! Is this something really suitable for using in the context of large-scale models that have already a lot of parameters and face with scarce and low quality observations particularly in terms of human-water interactions? Because of being heavily over-parameterized, this scheme is only suitable where there are at least multiple years of continuous and high quality data available: Even in North America, such data availability is widely limited considering the discontinuity in in-situ measurements of storage and release across regional reservoir networks even in western Canada and US, where most of the case studies of this work are located. The fact that many large dams are privately owned and therefore the data are not publicly available is not mentioned anywhere in the paper: This is the particularly the case of large hydroelectric dams in US, Canada and Brazil that together account for large proportion of annual reservoir storage globally. Please discuss properly this important issue of the scheme along with other limitations of the proposed model at least with the same weight as its strengths. Highlighting the limitation of the proposed algorithm must be a key consideration during revisions.

We agree with the reviewer that the DZTR scheme become over parameterized. However, its parameters are external to those of land surface model and are determined a priori from storage and release data. The decision of the time scale to use for specifying the parameters is left to the modeller. Given that storage (or level) and outflow data are required to calculate the parameters, the modeller will have the ability to see the seasonal patterns of the data and decide whether a monthly or a coarser time scale (e.g. quarterly) would be sufficient. The scheme is flexible in that regard. On the data issue, the actual operation rules are not usually known for privately owned reservoirs and that is why we are inferring them from release and storage data. Releases can be estimated from the nearest downstream station if direct releases are not available. Water levels of reservoirs are widely available and can be converted to storage data using the level volume relationship if known or a generalized one if not known (Liebe et al., 2005). If the data to parameterize the scheme cannot be found or reasonably estimated, then a simpler scheme like Hanaski et al. (2006) could be used.

The other point to keep in mind is that a land surface hydrology model can have several reservoir operation methods in parallel and use a reservoir identifiers as to which method to use for each reservoir. In a large scale modelling context, one would only consider reservoirs that have a considerable impact on the flow regime and those will tend to be larger important ones that have

reasonable flow and level records. As shown in Figure 4, we checked the reservoir locations in Canada and many of major ones have Water Survey Canada level records. Additionally, Alberta Environment and Parks make available such data for most reservoirs within Alberta (https://rivers.alberta.ca/). Similar high quality data is available for some basins in the US as well (Upper Colorado: https://www.usbr.gov/rsvrWater/HistoricalApp.html, Texas: https://www.waterdatafortexas.org/reservoirs/statewide). The manuscript is being revised to discuss those data issues and limitation to wider applicability of the DZTR scheme. Obtaining water level and flow data via satellites (Busker et al., 2019) may alleviate the issue in the future as discussed in the manuscript.

4) Section 4.1: What are the uncertainties in the generalized parameterizations? The percentiles corresponding to monthly target storage and release should be different for different reservoirs and I can imagine that it might be several combinations of percentiles that can provide similar modeling efficiency even in one single reservoir: Please discuss and provide some evidence on the uncertainty in these generalized parameterizations.

We agree with the reviewer that the uncertainty aspect has been overlooked in the manuscript. The following discussion points to the direction and scope we are following in the revised manuscript:

Reservoir operation on its own involves considerable uncertainties that is attributed to several factors. One major source of uncertainty in reservoir operation is future inflows (long-term and short-term inflow forecast). The forecast contains errors deep-rooted in the forecast method, the driving climate forecast, snowpack measurements, timing of snowmelt and the statistical (stationarity) assumptions to generate inflows based on historical inflows. The inflow forecast uncertainty is more significant during flood seasons because it involves subjective decisions of operators to averse the risk of dam overtopping and downstream flooding. Other sources of uncertainty in reservoir operation include changes in demand over time because of increases in demand for irrigation, power, water supply, etc. The purpose of the reservoir can also change from its initial intended purpose (e.g. adding a hydropower station to an irrigation dam). These changes are only implicitly captured by the DZTR scheme as implied in the storage and release time series used for parameterizing it for a specific reservoir.

Given the above uncertainties, even the actual reservoir operation may deviate from the designed reservoir operation rule curve. Some of the decisions of reservoir operators are spontaneous, ad-hoc, and depend on experiences that are not usually documented. Thus, there are difficulties to accurately represent the historical operation or to establish accurate relationships between reservoir storage, inflow, and release. These relationships typically contain considerable noise e.g., different release values at the same storage level during the same season. As a result, these uncertainties considerably influence the parameterization of the model derived to represent the reservoir

operation based on historical observations of each reservoir. This is particularly true for the algorithm presented because of two main factors. Firstly, the presented reservoir algorithm assumes that the relationship between reservoir storage and releases follow piecewise linear functions. There is a chance that other functional forms represent such relationship better for some reservoirs. Secondly, in the case of the generalized parameterization, the piecewise bending points (zone classification points) are estimated based on fixed probabilities of exceedance extracted from historical data for all reservoirs. A different dataset (of reservoirs and/or time periods) could result in different quantiles. The assumption of having similar bending points of the piecewise linear functions for all reservoirs cannot provide optimal zones for each reservoir. However, this is true for any type of generalization of parameters and we showed that the generalized parameterization performs better compared to other widely used algorithms. One way to reduce such type of uncertainty is to optimize the parameters based on observed data if it is available. Using all the optimal solutions usually encompasses the observed behaviour within a narrow bandwidth as shown in the example plot below. This model with parameterized using all the data available for Trinity reservoir in the US (Figure 1).

[Figure]

Figure 1: comparisons of Trinity dam simulation with generalized parametrization and multicriteria calibration

5) Figure 11 in the shows an explicit trade-off between reservoir release and reservoir storage during calibration: This means that it is impossible to reach the skill in representing each objective function without compromising on the other, implying that the algorithm is unable to track both

reservoir release and storage optimally at the same time: Isn't it a limitation in the model? How much this uncertainty contributes into uncertainty in identifying the role of reservoir in modifying the natural streamflow regime? This very important point seems to be wholly ignored at this stage and should be addressed in revisions.

Thanks for raising this point. This is being elaborated in the discussions in the revised manuscript as follows:

Only in the case of a perfect model and perfect data the trade-off between objectives converge to single point. The proposed model, like many other types of model is not an exception because of the uncertainties discussed in the previous point. Thus, the trade-off between storage and release objectives can be viewed as a measure of evaluate the limitation of the reservoir algorithm (piecewise linear functions, fixed number of zones, etc.) and observation errors. To examine the level of uncertainty of the trade-off, it is important to look at the shape and range of the trade-off on each objective function axis.

As shown in Figure 11 in the manuscript, except for few reservoirs, the value range of Pareto solutions for each objective function is generally narrow (check the axes) with good NSE values. In such cases the associated uncertainties are minimal and the trade-off between improving simulated releases and improving simulated storage is minimal. The figure above is a good example of this case, it shows how the simulations of reservoir release and storage using the parameter sets of the Pareto solutions enveloped the majority of the storage and release observations within a narrow uncertainty band. Conversely, in some cases, the extended spread of the tread-off in one of the axes (objective function) are observed, indicating a higher uncertainty of the algorithm or parameter sensitivity for the process the axis represent i.e. reservoir storage or release. This indicates further investigations of the datasets and parameterization for those reservoirs and their history of operations. Shifts in operational management of reservoirs do occur and these may obscure the parameterization. These may be detected by careful examination of the available records as well as metadata records of the reservoir history if accessible. The level of noise when determining the parameters could be an indicator of changes in operation.

Overall, given the good performance of the algorithm for almost all reservoirs using both the generalized and calibrated parameterization, it is suitable to simulate the effect of reservoir in modifying the natural flow regime with less uncertainty compared to other methods.

6) Figure 11 again: It is surprising that the results during validation do not show the trade-off observed during calibration in several reservoirs: Doesn't this show that the parametrization is very sensitive to the period used for parameter identification? Also, the results during calibration are non-dominated by definition; however, do the results during validation also remain nondominated when compared with other possible parametrizations that have been dominated during the calibration? The sensitivity of model parameters to training data and the robustness of results during validation should be well discussed during the revision and supported by experimental results.

Thanks for pointing this point out. We are incorporating this issue in the revised manuscript as:

Indeed, the calibration period used to identify the parameters influences the performance and shape of the Pareto front during validation period. One of the reasons the calibrated Pareto solution does not show the same trade-off during validation is when there is considerable change of inflow as a result of consecutive wet or dry years. As shown below (Figure 2) as an example for Glen Canyon (similarly Bhumibol, Fort Randall, Fort Peck), the calibration period has more wet and high inflow years than the validation period. Such considerable change of inflow, storage, and release results in performance failure during validation period.

[Figure]

Figure 2: Simulation of Glen Canyon dam with generalized parametrization

A small change of inflow, storage, or release in the validation period can change the shape of the trade-off, however, the calibrated parameters was still capable of reproducing good performance during validation close to or better than the generalized parameterization performance.

For the sake of demonstration, we calibrated the mentioned reservoirs using the whole observational record and all of them show the trade-off between storage and release fitting (please see the Figure 3 on the last page). Thus, we recommend using as much data as available to parameterize the model for a specific reservoir.

7) Incorporation of the algorithms in the considered large-scale model seems to be limited to one reservoir at the time. Whereas in real cases, multiple reservoirs are built over one river and therefore the cal/val procedure and the skill of the reservoir algorithm should be tested when the outflow from one reservoir is the inflow to the next reservoir. The paper ignores this as many other similar contributions do. But I believe this is worth at least proper discussion because the challenge

is out there and has remained, indeed, unsolved. Up to the time that the problem of considering multiple reservoirs in one basin is not properly solved, the results of large-scale models remain only as naive simulations of a virtual hydrologic reality at the basin-scale, which contributes to a huge uncertainty at regional, continental and global scales.

Thanks for pointing out this issue. It is indeed a major challenge to accurately represent reservoir operation for a cascade of reservoirs. The parametrization and formulation of the algorithm implicitly accounts, to some extent, for the upstream regulation effects from the upstream cascade reservoirs. This is because the regulated inflow is used for parametrizing downstream reservoirs. The regulated inflow is assumed to reflect the regulation of upstream reservoirs in the cascade. In reality, the operations of some cascade reservoirs are highly interlinked, particularly during the flood season. The decision regarding the release from one reservoir accounts for the (forecasted) state of other reservoirs. Such dual- or multi-linked operation is however not accurately accounted in the presented algorithm because it assumes that each reservoir operates using its own storage state, inflow and target storage and releases. Such systems require detailed modelling of operations that is not usually attainable in large scale hydrological models. Depending on the purpose of the model, the modeller may decide to lump those reservoirs together to improve simulations downstream. The issue raised is a general issue with the state of the art, and we agree that, as a community, we should look for innovative ways to handle the issue. In the revision, these discussion and clarification are being expounded.

References:

Busker, T., de Roo, A., Gelati, E., Schwatke, C., Adamovic, M., Bisselink, B., Pekel, J.-F. and Cottam, A.: A global lake and reservoir volume analysis using a surface water dataset and satellite altimetry, Hydrol. Earth Syst. Sci., 23(2), 669–690, doi:10.5194/hess-23-669-2019, 2019.

Liebe, J., van de Giesen, N. and Andreini, M.: Estimation of small reservoir storage capacities in a semi-arid environment: A case study in the Upper East Region of Ghana, Phys. Chem. Earth, Parts A/B/C, 30(6–7), 448–454, doi:10.1016/J.PCE.2005.06.011, 2005.

Lehner B, Liermann CR, Revenga C, Vörösmarty C, Fekete B, Crouzet P, Döll P, Endejan M, Frenken K, Magome J, et al. 2011. High-resolution mapping of the world's reservoirs and dams for sustainable river-flow management. *Frontiers in Ecology and the Environment* **9** (9): 494–502 DOI: 10.1890/100125

[Figure]

Figure 3: Calibration using the whole observational record

[Figure]

Figure 4: Reservoir water level stations of Water Survey Canada